# Bacterial peptidoglycan acts as a digestive signal mediating host adaptation to diverse food resources in *C. elegans*

Fanrui Hao[1,2], Huimin Liu[1,2] & Bin Qi [1] ✉

Food availability and usage is a major adaptive force for the successful survival of animals in nature, yet little is known about the specific signals that activate the host digestive system to allow for the consumption of varied foods. Here, by using a food digestion system in *C. elegans*, we discover that bacterial peptidoglycan (PGN) is a unique food signal that activates animals to digest inedible food. We identified that a glycosylated protein, Bacterial Colonization Factor-1 (BCF-1), in the gut interacts with bacterial PGN, leading to the inhibition of the mitochondrial unfolded protein response (UPR$^{mt}$) by regulating the release of Neuropeptide-Like Protein (NLP-3). Interestingly, activating UPR$^{mt}$ was found to hinder food digestion, which depends on the innate immune p38 MAPK/PMK-1 pathway. Conversely, inhibiting PMK-1 was able to alleviate digestion defects in *bcf-1* mutants. Furthermore, we demonstrate that animals with digestion defects experience reduced natural adaptation capabilities. This study reveals that PGN-BCF-1 interaction acts as "good-food signal" to promote food digestion and animal growth, which facilitates adaptation of the host animals by increasing ability to consume a wide range of foods in their natural environment.

Nematodes, which include bacterial-feeding nematodes, make up around 80% of all the multicellular animals on earth and play significant roles in many ecosystems[1]. In nature, nematode species are crucial for decomposing soil organic matter, improving soil aeration, mineralizing plant nutrients, and cycling nutrients[2]. One reason for their effectiveness is their remarkable capacity to consume a diverse array of food sources.

Food is vital for the survival and population regulation of animals, including nematodes, in natural habitats. Throughout evolution, food availability and preferences have played a significant role as an adaptive force[3]. To effectively obtain food, animals have developed sophisticated nervous system to sense food and guide their food intake[4–6]. After the consumption of food, it undergoes digestion in the gastrointestinal tract to extract essential nutrients for the animal[7]. A robust digestive system is crucial for animals to consume a wide variety of food sources, thereby enhancing their survival in challenging environments by increasing their ability to digest diverse foods. However, the specific signals from food that trigger nematodes to digest a broader range of foods in natural settings, as well as the signals that initiate the digestive process in animals, remain unclear. Identifying the common digestive signal and its underlying mechanism will provide insights into why bacterial-feeding nematodes thrive in nature and their critical roles in soil ecosystems.

The nematode *Caenorhabditis elegans* is a free-living roundworm that feeds on a variety of microbes environments rich in organic matter, such as rotting plant matter[8–10]. This species serves as an excellent model for studying food sensing and food-related behaviors[11]. Previously, we established a simple food digestion research system by investigating the effect of food digestion in *C. elegans* on development via feeding the inedible bacteria

[1]Southwest United Graduate School, Yunnan Key Laboratory of Cell Metabolism and Diseases, State Key Laboratory of Conservation and Utilization of Bioresources in Yunnan, Center for Life Sciences, School of Life Sciences, Yunnan University, Kunming, China. [2]These authors contributed equally: Fanrui Hao, Huimin Liu. ✉e-mail: qb@ynu.edu.cn

*Staphylococcus saprophyticus* (SS)[12,13]. *C. elegans* does not grow on either inedible food (*Staphylococcus saprophyticus*) or low-quality food (heat-killed *E. coli*, HK-*E. coli*), but does so when exposed to both[12]. Additionally, we have identified bacterial membrane proteins as signals from low-quality food (HK-*E. coli*) as well as the host's neural and innate immunity pathways that promote digestion of inedible food (*S. saprophyticus*)[12]. Given *C. elegans'* diet primarily consisting of various species of bacteria in nature, we propose that there may be a common signal from bacterial food that serves as a cue to activate the animal's digestive system. This activation could facilitate the breakdown of inedible food, thereby promoting the species' success in nature through enhanced digestion of a wider range of food sources.

In this paper, we discovered an unexpected role of bacterial peptidoglycan (PGN) as a signaling molecular that prompts host animals to digest inedible food. Mechanistically, we found that PGN was sensed by conserved intestinal glycosylated protein BCF-1 for inhibiting the mitochondrial unfolded protein response (UPR^mt) via neuropeptide, NLP-3, which promotes food digestion. Moreover, activation of UPR^mt inhibits food digestion, establishing a connection between the UPR^mt and digestion. Our findings collectively propose a model in which bacterial PGN functions as a signal for food digestion by triggering the process through a glycosylated protein that modulates intestinal UPR^mt via neuronal signals. This mechanism helps animals adapt to varying food conditions by facilitating efficient digestion.

## Results

### Low-quality food activates animals to digest inedible food, increasing their fitness in nature

Nematodes represent approximately 80% of all the multicellular animals on earth and play vital roles in various ecosystems[1]. The free-living nematode *Caenorhabditis elegans* is a widely used model organism in biological research. In nature, *C. elegans* thrives in microbe-rich environments, particularly in rooting plants, which serve as a significant food source for these organisms[8–10]. The ability of animals to digest a wide range of food sources is crucial for their survival and adaptation to their environment.

In our study, we categorized food into three distinct classes: good food, inedible food, and low-quality food (Fig.1a). We had previously established a simple research system to study food digestion in *C. elegans*[12,13], by observing the growth difference phenotype between in animals fed inedible *Staphylococcus saprophyticus* (SS) versus heat-killed *E. coli* (HK-*E. coli*) with SS together, which makes SS edible (Fig. 1a, b). Good food, such as laboratory *E. coli*-OP50, is easily digestible by animals and supports their growth (Fig. 1a). In contrast, inedible food (SS) cannot be digested and does not support growth (Fig. 1a, b). The low-quality food (HK-*E. coli*)[14], does not support animal growth because it lacks nutrition (Fig. 1a, b), however, the body size on HK-*E. coli* is bigger than on SS feeding, suggesting that animals can still digest and use HK-*E. coli* as nutrition (Fig. 1b). When low-quality food is mixed with inedible food, the HK-*E. coli* activates the food digestive system of the animals, and they are able to digest SS and grow (Fig. 1a, b). In natural environments, animals may not always have access to high-quality food. By consuming low-quality food, which activates their digestive system and enables them to digest more inedible food, animals can increase the diversity of their diet and enhance their survival in nature. Our findings suggest that animals possess a mechanism through which low-quality food triggers the activation of their digestive system, ultimately enhancing their fitness in their environment.

### Bacterial peptidoglycan activates animals to digest SS food

We have shown that HK-*E. coli* activates animals to digest SS[12]. We next screened for *E. coli* mutants that would be unable to promote animals to digest SS and support growth (Supplementary Fig. 1a). After screening, we discovered that animals exhibited significantly slower growth when fed on HK-*E. coli* mutants (*ΔycbB* and *ΔygeR*)+SS (Fig. 1c),

indicating a reduced capacity for food digestion in animals feeding these specific HK-*E. coli* mutants. YcbB is known to possess L, D-transpeptidase activity involved in peptidoglycan (PGN) maturation, while YgeR is presumed to play a role in PGN hydrolysis[15]. Therefore, we asked whether *E. coli* PGN itself could induce worms to digest SS by supplementing extracted PGN to SS. Remarkably, we found that the suppressed growth phenotype observed in worms fed solely on SS was alleviated when PGN was added (Fig. 1d), suggesting that PGN plays a pivotal role in activating food digestion in worms. In a previous study, we noted that *C. elegans* exhibited bloating in the intestinal lumen when fed on living SS, indicating an inability to digest this food source effectively[12].To assess the effectiveness of PGN in promoting food digestion, we measured the width of the intestinal lumen in animals and observed that the bloating was reduced in animals supplemented with PGN (Fig. 1e). These findings suggest that PGN indeed facilitates the digestion of SS in animals.

Peptidoglycan (PGN), which is a vital component of bacterial cell walls and is composed of β(1–4)-linked N-acetylglucosamine (GlcNAc) and N-acetylmuramic acid (MurNAc) crosslinked by short peptides, may play a crucial role in activating food digestion in animals. To examine whether the effects of bacterial PGN on animal growth are bacteria-specific, we evaluated the impact of PGN extracted from different bacterial species on animal growth. Specifically, we selected *Enterococcus faecalis (E. f)*, an inedible food that cannot support animal growth adequately[12], and *Bacillus subtilis (B. s)*, a beneficial food that enhances the lifespan of *C. elegans*[16,17]. We observed that PGN extracted from both *E. coli* and *B. s* species can promote animal digestion of SS to support growth, whereas PGN extracted from *E. f* did not have the same effect (Supplementary Fig. 1b). This finding suggests that the effects of bacterial PGN on animal growth may be strain-specific.

Next, to identify the specific unit of PGN that activates food digestion in animals, we used several enzymes that cleave PGN at specific sites (Fig.1f). We found that PGN treated with N-acetylmuramoyl-L-alanine amidase AmiD, Beta-hexosaminidase NagZ (Supplementary Fig. 1c) or ProteinaseK could no longer activated the animals to consume SS (Fig.1d). However, lysozyme treated PGN still promoted animals to consume SS for supporting growth (Fig. 1d). These results suggest that the specific PGN molecules effective in activating food digestion are 5'NAG-NAM disaccharide muropeptides with an amino acid peptide attached to NAM.

### Screen for PGN-binding proteins involved in food digestion

In order to determine how PGN activates food digestion, we began by investigating whether *E. coli*-binding proteins[18] and PGN-binding proteins[15] have a role in food digestion. 44 proteins were identified as both *E. coli*-binding and PGN-binding (Fig. 2a). Among the identified 44 proteins that bind to both *E. coli* and PGN, 23 genes were found to be expressed in the intestine (Fig. 2a), the primary digestive organ in animals. If the particular PGN-interaction protein is critically involved in food digestion, then RNAi knockdown of this candidate may generate a slow growth phenotype in animals fed HK-*E. coli* + SS. We, therefore, performed an RNAi screen of all 23 candidate genes (Supplementary Fig. 2) and found that animals fed with *bcf-1* RNAi exhibited slow growth when fed with HK-*E. coli* + SS (Fig. 2b). Additionally, two independent *bcf-1* mutant strains also displayed slow growth when exposed to HK-*E. coli* + SS (Fig. 2c). Moreover, the intestinal lumen of *bcf-1* mutant animals remained distended when fed HK-*E. coli* + SS (Fig. 2d). These findings indicate that the PGN-binding protein BCF-1 is involved in facilitating the digestion of SS food in animals.

### PGN interacts with BCF-1 to facilitate food digestion

Our previous research revealed that BCF-1, an intestinal N-glycosylated protein, plays a role in regulating *E. coli* colonization by directly binding to the bacteria[18]. In the HK-*E. coli* + SS feeding condition, BCF-1 is only expressed in the intestine (Fig. 3a). Animals with intestinal *bcf-1*

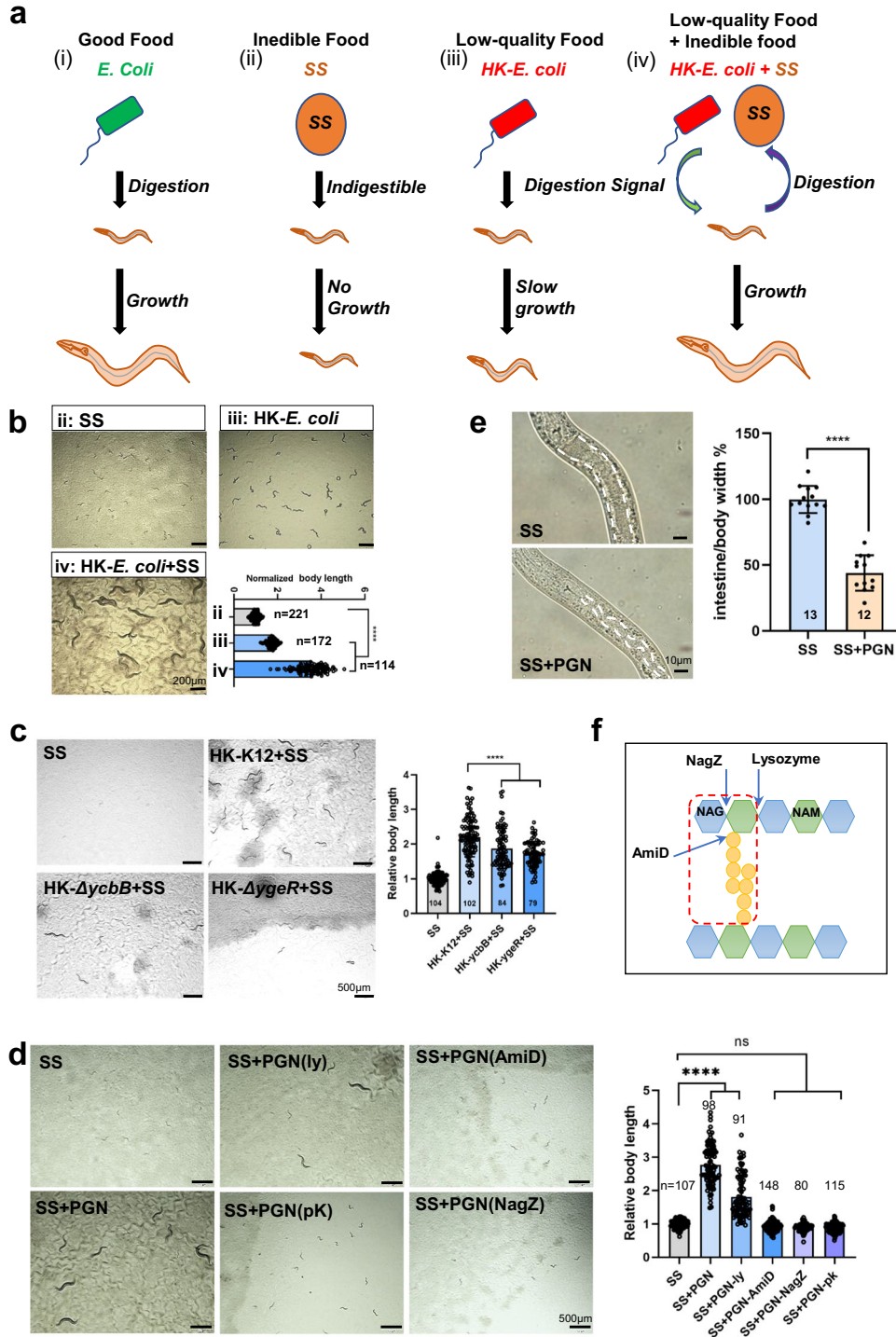

RNAi exhibited slow growth when fed HK-*E. coli* + SS (Fig. 3b), suggesting that intestinal BCF-1 is involved in activating food digestion.

We then investigated whether PGN-triggered food digestion is dependent on its interaction with the BCF-1 protein. First, we incubated PGN with worm lysate (BCF-1::FLAG) and found that PGN bound to BCF-1 in a concentration-depended manner (Fig. 3c), suggesting that PGN interacts with BCF-1. Secondly, we treated PGN with proteinase K and found that proteinase K-treated PGN cannot activate animals to digest SS (Fig. 1d). Moreover, proteinase K-treated PGN failed to pull-down BCF-1 (Fig. 3d), suggesting that the short amino acid peptides in PGN are essential for binding to BCF-1. Thirdly, we found that BCF-1 expression is induced by PGN in the SS food feeding condition (Fig. 3e, f), implying that PGN activates food digestion through enhancing BCF-1 expression.

Finally, *bcf-1* mutants exhibited slow growth on SS + PGN (Fig. 3g), indicating that PGN promotes SS digestion which requires BCF-1. Together, these results indicated that PGN activates food digestion through its interaction with the BCF-1 protein in the intestine.

Moreover, *C. elegans* BCF-1 is a conserved glycosylated protein found only in bacterial-feeding nematodes (Supplementary Fig. 3a), including in *C. remanei, C. nigoni, C. briggsae, C. bovis,* and *C. auriculariae*. This indicates that conserved nematode BCF-1 may sense unique bacterial PGN to activate food digestion in worms, which facilitates adaptation of the nematodes in their natural environment by increasing their ability to consume a wide range of bacterial foods.

In order to assess whether the influence of BCF-1 on food digestion is solely through PGN interaction, we performed an animal growth

**Fig. 1 | PGN activates *C. elegans* digestive system. a** The cartoon illustration shows the developmental progression of animals grown on different quality foods. The signal from low-quality food (HK-*E. coli*) activates animals to digest inedible food, *Staphylococcus saprophyticus* (SS), which potentially increases fitness in animals by enabling them to consume a wider range of foods in nature. **b** Developmental phenotype of N2 grown on the SS, HK-*E. coli*, and HK-*E. coli* + SS at 20°C for 7d. *n* = 221 for SS, *n* = 172 for HK-*E. coli*, *n* = 114 for HK-*E. coli* + SS. Statistical significance was calculated using multiple unpaired *t*-tests (two-tailed). Obtained *p* values were as follows: SS vs. HK-*E. coli*; ****p* < 0.0001. SS vs. HK-*E. coli* + SS; ****p* < 0.0001. Scale bar, 500μm. **c** Developmental phenotype of N2 grown on the HK-*E. coli* (*ΔycbB* or *ΔygeR* mutant) + SS or SS at 20°C for 5d post-L1 synchronization. *n* = 104 for SS, *n* = 102 for HK-K12 + SS, *n* = 84 for HK-*ycbB* + SS, *n* = 79 for HK-*ygeR* + SS. Statistical significance was calculated using multiple unpaired *t*-tests (two-tailed). Obtained *p* values were as follows: HK-K12 + SS vs. HK-*ycbB* + SS and HK-*ygeR* + SS; ****p* < 0.0001, respectively. Scale bar, 500μm **d** Developmental progression of animals grown on SS+ enzyme-treated PGN at 7d at 20°C. Scale bar, 500μm. ly: lysozyme; pK: proteinaseK; NagZ, glucosaminidase; AmiD, amidase; *n* = 107 for SS, *n* = 98 for SS + PGN, *n* = 91 for SS + PGN-ly(lysozyme), *n* = 148 for SS + PGN-AmiD, *n* = 80 for SS + PGN-NagZ, *n* = 115 for SS + PGN-pk(proteaseK). Statistical significance was calculated using multiple unpaired *t*-tests (two-tailed). Obtained *p* values were as follows:SS vs. SS + PGN and SS + PGN-ly(lysozyme); ****p* < 0.0001, respectively. **e** Microscope image and bar graph show the relative width of intestinal lumen in N2 animals fed SS or SS + PGN food. *n* = 13 for SS, *n* = 12 for SS + PGN. Statistical significance was calculated using multiple unpaired *t*-tests (two-tailed). Obtained *p* values were as follows: SS vs. SS + PGN; ****p* < 0.0001. **f** Schematic representation of the peptidoglycan structure and cleavage points of enzymes by arrows. Red box indicates the structure for the digestion signal from PGN, which contains 5'NAG-NAM disaccharide muropeptides with an amino acid peptide attached to NAM. NagZ, glucosaminidase; AmiD, amidase; NAG, 3N'-acetylglucosamine; NAM, 5N'-acetylmuramic acid. The developmental progression of animals is scored by relative worm body length. For all panels, *n*= number of animals which were scored from at least three independent experiments. Data are represented as mean ± SD. ****p* < 0.0001, *** *P* < 0.001, ***P* < 0.01, **P* < 0.05, ns: no significant difference. Source data are provided as a Source Data file.

assay. We observed that in the absence of PGN (HK-*ΔycbB* + SS), animal growth was significantly impaired (Supplementary Fig. 3b). Conversely, the presence of PGN-containing HK-*E. coli* facilitated animals in digesting SS for growth. Intriguingly, HK-*E. coli* was unable to support SS digestion in *bcf-1* mutant animals (Supplementary Fig. 3b), suggesting that HK-*E. coli* promotes digestion through the action of the PGN-binding protein BCF-1. Furthermore, we noted that the impact of the *bcf-1* mutation on food digestion, specifically SS digestion, was substantially greater compared to the effect of PGN deficiency (*ΔycbB*) (Supplementary Fig. 3b). These findings hint at the possibility of an additional, BCF-1-independent effect of PGN on food digestion, warranting further exploration in future studies.

## PGN inhibits UPR^mt through BCF-1 for food digestion

Bacterial peptidoglycan muropeptides have been shown to promote mitochondrial homeostasis by acting as ATP synthase agonizts and inhibiting mitochondrial unfolded protein response (UPR^mt)[15]. In our study, we also observed induction of UPR^mt in worms fed with the PGN mutant*ΔycbB* (Fig. 4a). This led us to investigate whether PGN promotes food digestion by inhibiting UPR^mt in conjunction with BCF-1. It was hypothesized that if the PGN-binding protein BCF-1 is involved in the beneficial effects of PGN on mitochondrial homeostasis, mutation of *bcf-1* may lead to phenotype similar to those caused by a diet lacking PGN, resulting in UPR^mt induction and increased food avoidance behavior[15]. Consistent with this hypothesis, we observed UPR^mt induction in animals with *bcf-1* RNAi (Fig. 4b) or a *bcf-1* mutation (Fig. 4a), along with increased food avoidance behavior in the *bcf-1* mutant (Fig. 4c), indicating similarities in phenotype between BCF-1 mutation and PGN deficiency.

Next, we asked whether PGN has role in inhibiting UPR^mt through BCF-1. We found that supplementation of PGN to animals fed with *ΔycbB* suppressed UPR^mt, with this effect being abolished in *bcf-1* mutant worms (Fig. 4a), (Fig. 4a), suggesting that PGN-BCF-1 interaction plays a critical role in maintaining the UPR^mt. However, the impact of feeding *ΔycbB* bacteria on *hsp-6* expression was more pronounced compared to that of *bcf-1(-)* mutant worms, indicating a potential BCF-1-independent effect of PGN on UPR^mt. Previous research[15] have demonstrated that PGN can inhibit UPR^mt through its binding to ATP synthase, leading to increased ATP synthase activity. Thus, there is a possibility of PGN-BCF-1 and PGN-ATP synthase interactions in the regulation of UPR^mt.

We then asked whether PGN has a function in activating food digestion and promoting animals' growth through BCF-1. Our findings showed that supplementation of PGN to animals fed with *ΔycbB* resulted in enhanced animal growth, which was negated in *bcf-1* mutant worms (Supplementary Fig. 4), indicating the importance of PGN-BCF-1 interaction in activating food digestion and promoting animal growth.

Finally, we wondered if a balanced UPR^mt is critical for food digestion. ATFS-1, the key factor in the UPR^mt pathway, is sufficient to activate UPR^mt. We found that the *afts-1(et18, gain of function)* mutant, which has constitutive activation of UPR^mt, grew significantly slower on HK-*E. coli* + SS (Fig. 4d), suggesting that activation of UPR^mt inhibits food digestion. *bcf-1* mutants show a decreased ability to digest food (Fig. 2b, c) and increased UPR^mt (Fig. 4a, b). Thus, we speculated that activation of UPR^mt in *bcf-1* mutant may decrease the ability to digest food in animals. Moreover, we found that digestion defects in *bcf-1* RNAi animals was recovered by mutation of *atfs-1*, which abolished the UPR^mt activation (Fig. 4e).

In conclusion, our data highlight the critical role of PGN-BCF-1 interaction in activating food digestion, partially through the inhibition of UPR^mt.

## *bcf-1* mutation activates UPR^mt through a cell-non-autonomous mechanism, dependent on neuropeptide NLP-3

Next, we wondered how *bcf-1* mutation triggers the UPR^mt activation. Key transcription factors such as DVE-1, ATFS-1, and UBL-5 are known to be involved in the UPR^mt pathway. We found that DVE-1 is translocated into the nucleus in the animals with *bcf-1* RNAi (Fig. 5a) and RNAi these key factors could suppress *bcf-1* mutation induced UPR^mt (Fig. 5b), suggesting that the UPR^mt activation caused by *bcf-1* mutation is dependent on the canonical pathway of UPR^mt.

Then, we analyzed the RNA-seq data of *bcf-1* mutant[18] and found that up-regulated genes involved in neuropeptide signaling pathway are enriched (Supplementary Fig. 5a), with neuropeptide genes showing up-regulation in *bcf-1* mutant (Supplementary Fig. 5b). Neuropeptides are known to regulate cell nonautonomous UPR^mt,[19–21], particularly from neurons to intestine. This led us to explore whether intestinal *bcf-1* regulates UPR^mt through neuronal signaling. We examined whether *bcf-1* dysfunction-induced UPR^mt is dependent on neuroendocrine secretion. We found that UPR^mt activation in *bcf-1 (ok2599)* mutant animals were reduced by knock-down of *egl-3* and *egl-21* (Fig. 5c) which are involved in neuropeptide processing. In addition, through screening neuropeptide genes up-regulated in *bcf-1* mutants (Supplementary Fig. 5b), we identified that knock-down of *nlp-3* inhibited the activated UPR^mt in *bcf-1 (ok2599)* animals (Fig. 5d).

Under HK-*E. coli* + SS feeding condition, we observed digestion defects and developmental delay in *bcf-1 (ok2599)* mutants. However, we found that these defects were partially restored upon RNAi targeting *nlp-3*, *egl-3*, and *egl-21* (Fig. 5e). Although this partial rescue indicated the involvement of *nlp-3*, there were still residual defects observed, suggesting the presence of other factors, in addition to *nlp-3*, that play a role in BCF-1-mediated regulation of food digestion. We also observed that knocking down *nlp-3* reduced the food avoidance response induced by the *bcf-1* mutation (Supplementary Fig. 5c),

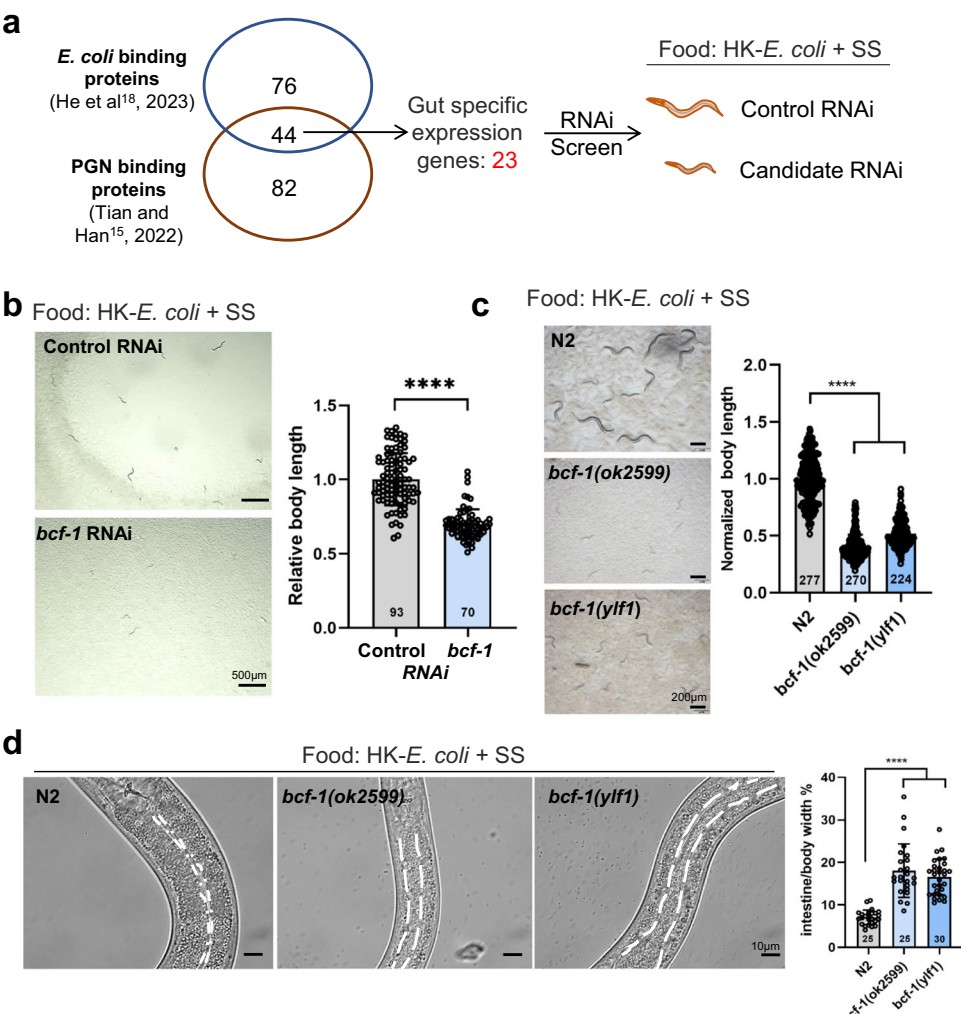

**Fig. 2 | Screening PGN-Binding proteins involved in food digestion. a** Cartoon illustration showing the screening strategy used to find the PGN-binding protein that activates animals to digest SS. Venn diagram shows the total number of identified *E. coli* binding proteins[18] and PGN binding proteins[15], along with their overlap. Out of the 44 overlapping genes, 23 intestinal-specific genes were tested by RNAi screening. RNAi knockdown of the candidate genes is expected to result in a slow-growth phenotype in animals fed HK-*E. coli* + SS. **b, c** Developmental progression of animals with *bcf-1* RNAi (**b**) or *bcf-1* mutations (**c**) when grown on HK- *E. coli* + SS. **b** $n = 93$ for control RNAi, $n = 70$ for *bcf-1* RNAi. Statistical significance was calculated using multiple unpaired *t*-tests (two-tailed). Obtained *p* values were as follows: control RNAi vs. *bcf-1* RNAi; ****$p < 0.0001$. **c** $n = 277$ for N2, $n = 270$ for *bcf-1(ok2599)*, $n = 224$ for *bcf-1(ylf1)*. Statistical significance was calculated using multiple unpaired *t*-tests (two-tailed). Obtained *p* values were as follows: N2 vs. *bcf-1(ok2599)* and *bcf-1(ylf1)*; ****$p < 0.0001$, respectively. **d** Microscope images and bar graph show the relative width of the intestinal lumen between N2 and *bcf-1* mutants when fed HK-*E. coli* + SS. $n = 25$ for N2, $n = 25$ for *bcf-1(ok2599)*, $n = 30$ for *bcf-1(ylf1)*. Statistical significance was calculated using multiple unpaired *t*-tests (two-tailed). Obtained *p* values were as follows: N2 vs. *bcf-1(ok2599)* and *bcf-1(ylf1)*; ****$p < 0.0001$, respectively. Developmental progression of animals is scored by relative worm body length. For all panels, *n*= number of animals which were scored from at least three independent experiments. Data are represented as mean ± SD. ****$p < 0.0001$, *** $P < 0.001$, **$P < 0.01$, *$P < 0.05$, ns: no significant difference. Source data are provided as a Source Data file.

indicating that the food avoidance behavior in *bcf-1* mutants is also dependent on NLP-3-mediated UPR^mt activation.

In summary, our findings indicate that *bcf-1* mutation triggers UPR^mt activation by modulating the release of the neuropeptide NLP-3, leading to inhibition of food digestion and food avoidance behavior. Additionally, our results suggest that intestinal signaling may transmit signals to neurons to inhibit neuropeptide release, thereby activating food digestion through the inhibition of intestinal UPR^mt.

### Inhibition of innate immunity rescues the digestive defects of *bcf-1(lf)* mutants

Our previous study demonstrated that inhibition of innate immune PMK-1 pathway activates digestion[12]. In this study, we observed elevated levels of phosphorylated PMK-1 (p-PMK-1) in *bcf-1* mutant worms when fed with *E. coli*-OP50 (Fig. 6a), suggesting that the activation of the PMK-1 pathway may contribute to the digestive defects in *bcf-1*

mutants. Consistent with this, we found increased p-PMK-1 levels in *bcf-1* mutants under HK-*E. coli* + SS feeding conditions (Fig. 6b). To further examine the role of the PMK-1 pathway in *bcf-1* mutant digestive defects, we generated double mutants *bcf-1(ok2599);pmk-1(km25)* and observed that the food digestion defects (developmental delay) in *bcf-1* mutants were rescued in the double mutant (Fig. 6c). Additionally, the food avoidance response induced by the *bcf-1* mutation was reduced in the *bcf-1,pmk-1* double mutants (Supplementary Fig. 6), indicating that the food avoidance behavior in *bcf-1* mutants indeed depends on PMK-1. These results suggest that *bcf-1* mutation activates innate immunity through the PMK-1 pathway, leading to inhibition of food digestion and increased food avoidance.

The *atfs-1* (*et18*, gain of function) mutant decreases the ability to digest food by continuous activation of UPR^mt in (Fig. 4d). We then wondered whether inhibition of digestion by UPR^mt activation is dependent on PMK-1. Our analysis revealed an upregulation of p-PMK-1

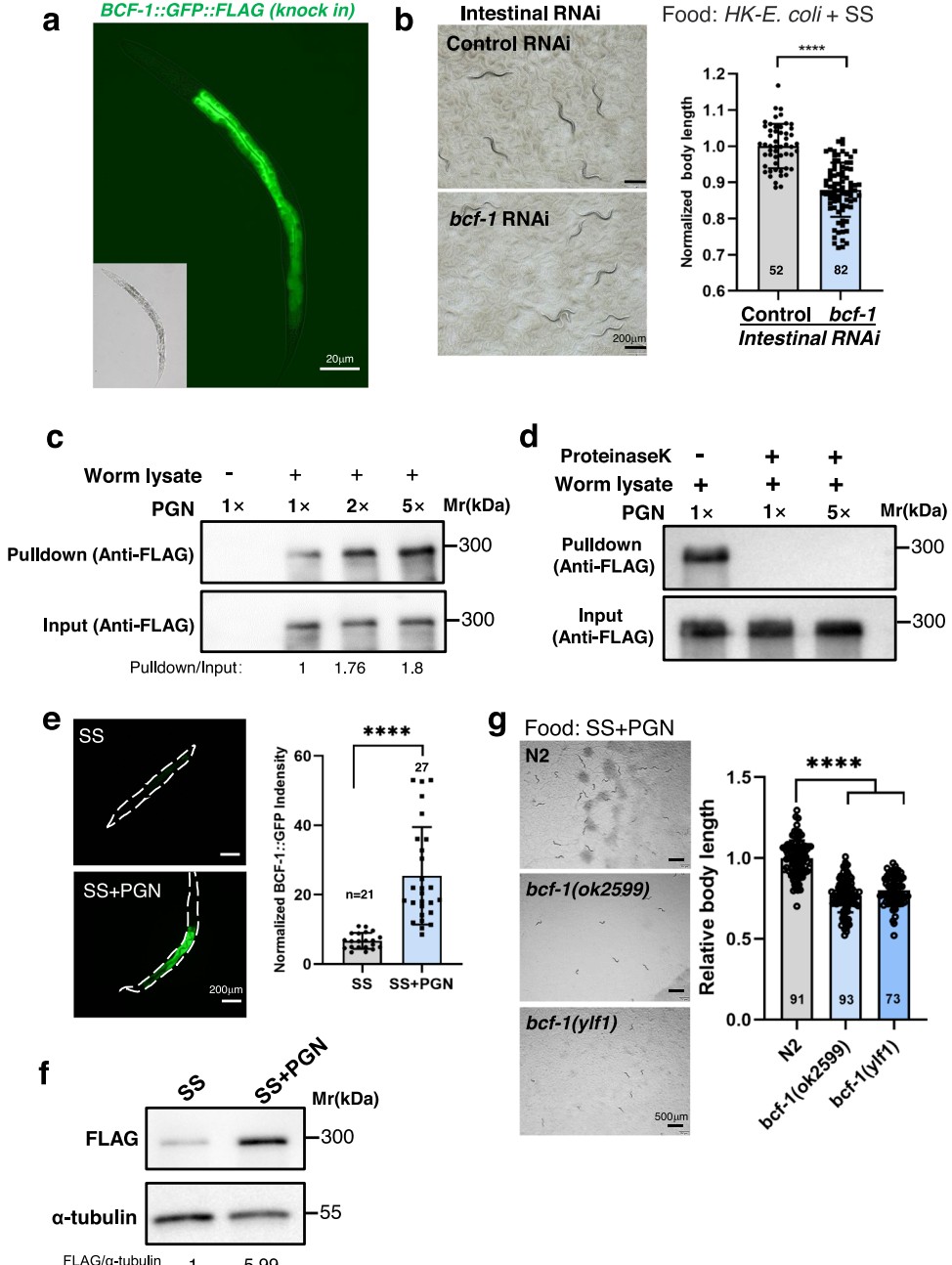

**Fig. 3 | PGN activates food digestion through BCF-1. a** Fluorescence image showing that BCF-1 is specifically expressed in the intestine, using a single-copy insertion of the *bcf-1p::bcf-1::gfp::flag* reporter strain. **b** Developmental progression of animals with intestinal-specific *bcf-1* RNAi when grown on HK- *E. coli* + SS for 4d at 20 °C. *n* = 52 for control RNAi, *n* = 82 for *bcf-1* RNAi.Statistical significance was calculated using multiple unpaired *t*-tests (two-tailed). Obtained *p* values were as follows: control RNAi vs. *bcf-1* RNAi; ****p < 0.0001. **c** In vitro PGN binding assay (pull-down assay) showing that PGN interacts with BCF-1 protein. BCF-1 from worm lysates (*bcf-1p::bcf-1::gfp::flag* reporter strain) was bound to PGN and the binding increased in a concentration-dependent manner. Anti-Flag was used as an input control, which indicate that PGN was incubated with an all most equivalent amount of BCF-1 tagged (BCF-1-GFP-FLAG) proteins. **d** Proteinase K treatment, which is sensitive to short peptides on PGN, eliminated the binding of PGN to BCF-1. **e** Microscope images and bar graph showing that BCF-1::GFP expression is induced

in the *bcf-1p::bcf-1::gfp::flag* reporter strain when fed with SS + PGN. *n* = 21 for SS, *n* = 27 for SS + PGN. Statistical significance was calculated using multiple unpaired *t*-tests (two-tailed). Obtained *p* values were as follows: SS vs. SS + PGN; ****p < 0.0001. **f** Western blot showing the level of BCF-1::GFP::FLAG in L1 animals feeding SS and SS + PGN. **g** Developmental progression of the *bcf-1* mutant grown on SS + PGN at 5d at 20 °C. *n* = 91 for N2, *n* = 93 for *bcf-1(ok2599)*, *n* = 73 for *bcf-1(ylf1)*. Statistical significance was calculated using multiple unpaired *t*-tests (two-tailed). Obtained *p* values were as follows: N2 vs. *bcf-1(ok2599)* and *bcf-1(ylf1)*; ****p < 0.0001, respectively. The developmental progression of animals is scored by relative worm body length. For all panels, *n* = number of animals that were scored from at least three independent experiments. Data are represented as mean ± SD. ****p < 0.0001, *** P < 0.001, **P < 0.01, *P < 0.05, ns: no significant difference. Source data are provided as a Source Data file.

levels in *atfs-1(et18)* mutant animals with constitutive UPR$^{mt}$ activation (Fig. 6d), suggesting a positive regulatory role of UPR$^{mt}$ activation on PMK-1. However, when examining the expression of *hsp-6*, a UPR$^{mt}$ marker, we found that PMK-1 did not significantly impact UPR$^{mt}$

activation (Fig. 6e). In the animal growth assay, we observed that UPR$^{mt}$ activation inhibited animal growth in wild-type N2 and *pmk-1* mutant backgrounds (Fig. 6f), indicating a negative effect on food digestion. Remarkably, the slow growth phenotype observed in *atfs-1(et18)*

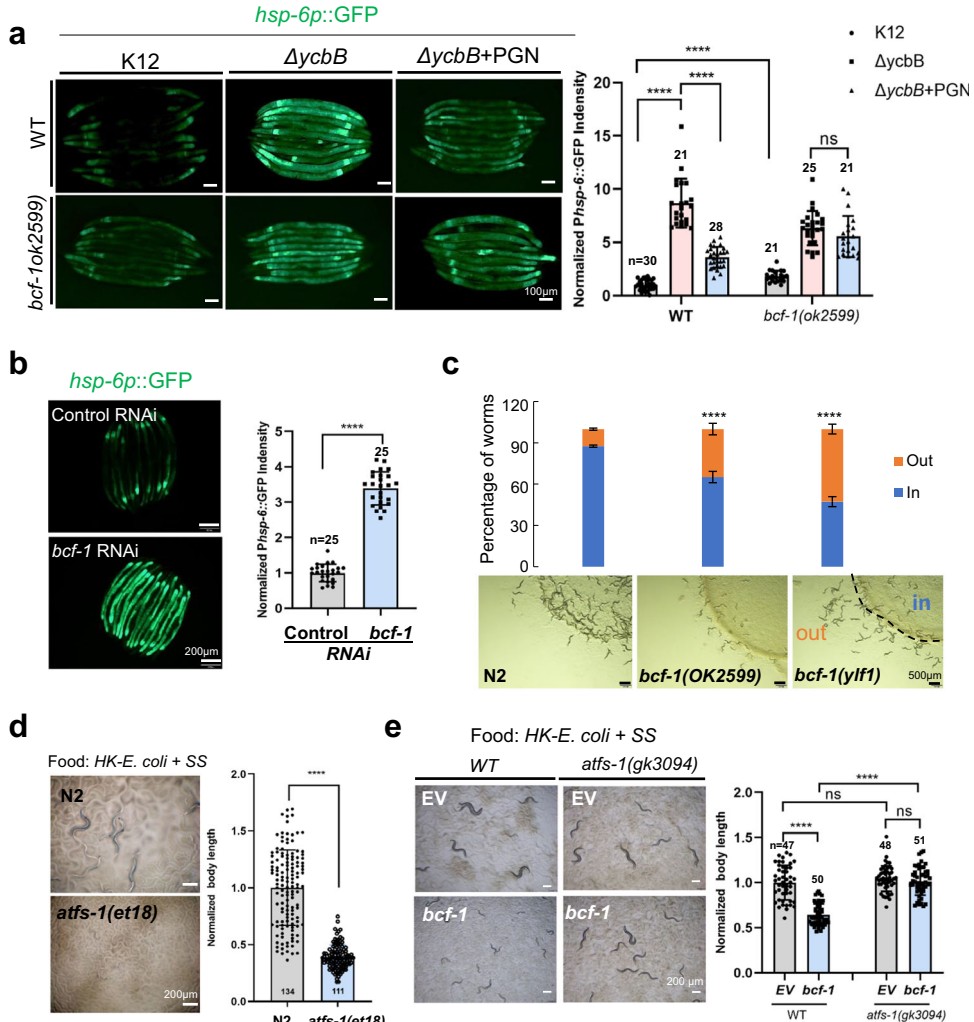

**Fig. 4 | PGN inhibits UPRmt through *bcf-1* for food digestion. a** Microscope image and bar graph showing the role of PGN in inhibiting UPRmt depends on BCF-1. UPRmt was activated by feeding the PGN mutant (*ΔycbB*), and was suppressed by the addition of PGN. However, the addition of PGN failed to suppress UPRmt in the *bcf-1* mutant. $n$ = 30, 22 and 28 for WT on K12, *ΔycbB* and *ΔycbB* + PGN, respectively. $n$ = 21, 25 and 21 for *bcf-1(ok2599)* on K12, *ΔycbB* and *ΔycbB* + PGN, respectively. Statistical significance was calculated using multiple unpaired *t*-tests (two-tailed). Obtained $p$ values were as follows: WT on K12 vs. WT on *ΔycbB* and *ΔycbB* + PGN; ****$p < 0.0001$, respectively. *bcf-1(ok2599)* on K12 vs. *bcf-1(ok2599)* on *ΔycbB* and *ΔycbB* + PGN; ****$p < 0.0001$, respectively. WT on K12 vs. *bcf-1(ok2599)* on K12; ****$p < 0.0001$. *bcf-1(ok2599)* on *ΔycbB* vs. *bcf-1(ok2599)* on *ΔycbB* + PGN; $p = 0.1164$. **b** Fluorescence image and bar graph show that UPRmt was induced in animals with *bcf-1* RNAi treatment. $n$ = 25 for control and *bcf-1* RNAi Statistical significance was calculated using multiple unpaired *t*-tests (two-tailed). Obtained $p$ values were as follows: control vs. *bcf-1* RNAi; ****$p < 0.0001$. **c** The food avoidance phenotype of *bcf-1* mutants is depicted. Food avoidance is increased in animals with the *bcf-1*

mutation. 200–400 animals/assay, mean ± SD from 3 replicates. Obtained $p$ values were as follows: N2 vs. *bcf-1(ok2599)* and *bcf-1(ylf1)*; ****$p < 0.0001$, respectively. **d** Developmental progression of *atfs-1(gf)* grown on HK- *E. coli* + SS at 25℃ for 3d. $n$ = 134 for N2, $n$ = 111 for *atfs-1(et18)*. Statistical significance was calculated using multiple unpaired *t*-tests (two-tailed). Obtained $p$ values were as follows: N2 vs. *atfs-1(et18)*; ****$p < 0.0001$. **e** Development progression of *atfs-1(gk3094)* mutant animals with *bcf-1* RNAi grown on HK- *E. coli* + SS at 25℃ for 3d. $n$ = 47 and 50 for WT on EV and *bcf-1* RNAi, $n$ = 48 and 51 for *atfs-1(gk3094)* on EV and *bcf-1* RNAi. Statistical significance was calculated using multiple unpaired *t*-tests (two-tailed). Obtained $p$ values were as follows: WT on EV vs. WT on *bcf-1* RNAi; ****$p < 0.0001$. WT vs. *atfs-1(gk3094)* on *bcf-1* RNAi; ****$p < 0.0001$. WT vs. *atfs-1(gk3094)* on EV; $p = 0.2049$. *atfs-1(gk3094)* on EV vs. *atfs-1(gk3094)* on *bcf-1* RNAi; $p = 0.3279$. For all panels, $n$ = number of animals which were scored from at least three independent experiments. Data are represented as mean ± SD. ****$p < 0.0001$, *** $P < 0.001$, **$P < 0.01$, *$P < 0.05$, ns: no significant difference. Source data are provided as a Source Data file.

mutant animals was partially rescued by *pmk-1* mutation (Fig. 6f), indicating that PMK-1 contributes partly to the inhibitory effect of UPRmt activation on food digestion. Furthermore, our results suggest the involvement of other PMK-1-independent pathways in the inhibition of food digestion during UPRmt activation.

### The PGN-BCF1 signaling system serves as a "good-food signal" that promotes animal adaptation
Food intake and choices play a crucial role in the survival of animals in their natural habitats[3]. Our research indicates that low-quality food (HK- *E. coli*) triggers animals to consume inedible food, thereby

increasing the diversity of available food sources (Fig. 1), potentially enhancing their fitness by supporting a larger population in their environment. We also present evidence highlighting the significance of the PGN-BCF1 signaling pathway in enabling animals to digest inedible food like SS.

To further explore the contribution of the PGN-BCF1 signaling system to the adaptation of bacteria-eating worms by enhancing their capacity to consume a variety of food sources in their natural environment, we evaluated the population size of animals under different feeding conditions. We initiated experiments by introducing a single animal to various food conditions and monitored the total number of

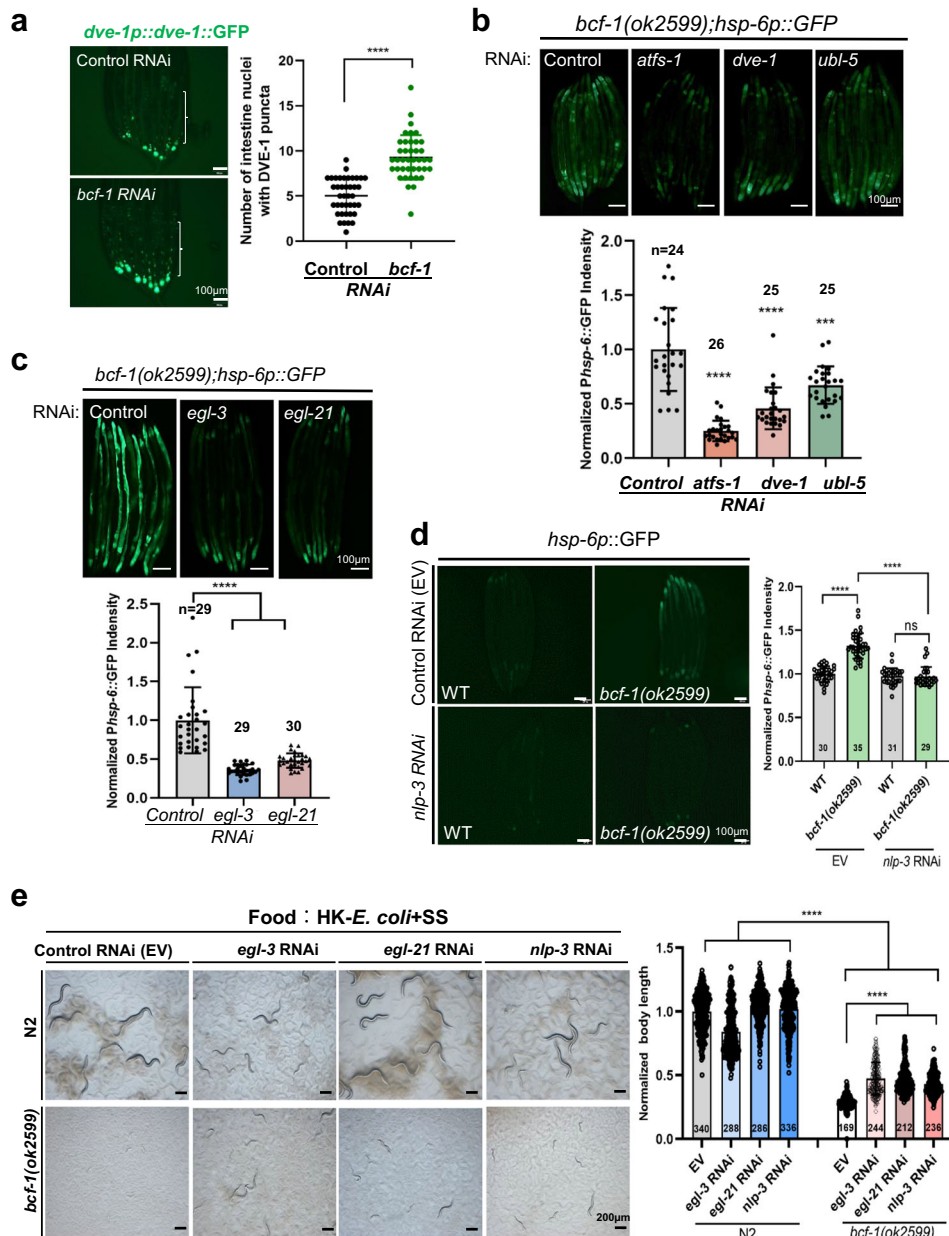

**Fig. 5 | UPR^mt activation by *bcf-1* mutation depends on neuropeptide NLP-3.**
**a** Fluorescence image and bar graph showing that DVE-1::GFP was accumulated in intestinal nuclei in *dve-1p::dve-1::gfp* animals with *bcf-1* RNAi treatment. *n* = 42 for control and *bcf-1* RNAi. Statistical significance was calculated using multiple unpaired *t*-tests (two-tailed). Obtained *p* values were as follows: control vs. *bcf-1* RNAi; ****p < 0.0001. **b** Fluorescence image and bar graph showing the level of *hsp-6p::GFP* in *bcf-1 (ok2599)* mutant animals with *atfs-1, dve-1* and *ubl-5* RNAi. *n* = 24 for control RNAi, *n* = 26 for *atfs-1* RNAi, *n* = 25 for *dve-1* RNAi, *n* = 25 for *ubl-5* RNAi. Statistical significance was calculated using multiple unpaired *t*-tests (two-tailed). Obtained *p* values were as follows: control vs. *atfs-1* RNAi, *dve-1* RNAi and *ubl-5* RNAi; ****p < 0.0001, respectively. **c** Fluorescence image and bar graph showing the level of *hsp-6p::GFP* in *bcf-1 (ok2599) mutant* animals with *egl-3* and *egl-21* RNAi. *n* = 29 for control RNAi, *n* = 29 for *egl-3* RNAi, *n* = 30 for *egl-21* RNAi. Statistical significance was calculated using multiple unpaired *t*-tests (two-tailed). Obtained *p* values were as follows: control vs. *egl-3* RNAi and *egl-21* RNAi; ****p < 0.0001, respectively. **d** Fluorescence image and bar graph showing the level of *hsp-6p::GFP*

in *bcf-1 (ok2599)* mutant animals with *nlp-3* RNAi. *n* = 30 and 31 for WT on control and *nlp-3* RNAi. *n* = 35 and 29 for *bcf-1(ok2599)* on control and *nlp-3* RNAi. Statistical significance was calculated using multiple unpaired *t*-tests (two-tailed). Obtained *p* values were as follows: WT on control vs. *nlp-3* RNAi ****p < 0.0001. *bcf-1(ok2599)* on WT vs. *nlp-3* RNAi ****p < 0.0001. WT vs. *bcf-1(ok2599)* on *nlp-3* RNAi; *p* = 0.8890. **e** Development progression of *bcf-1 (ok2599)* mutant animals with *nlp-3, egl-3* and *egl-21* RNAi when grown on HK- *E. coli* + SS at 20℃ for 4d. *n* = 340, 288, 286 and 336 for N2 on EV, *egl-3* RNAi, *egl-21* RNAi and *nlp-3* RNAi. *n* = 169, 244, 212 and 236 for *bcf-1(ok2599)* on EV, *egl-3* RNAi, *egl-21* RNAi and *nlp-3* RNAi. Statistical significance was calculated using multiple unpaired *t*-tests (two-tailed). Obtained *p* values were as follows: N2 vs. *bcf-1(ok2599)*; ****p < 0.0001. *bcf-1(ok2599)* on EV vs. *egl-3* RNAi, *egl-21* RNAi and *nlp-3* RNAi; ****p < 0.0001, respectively. For all panels, n= number of animals which were scored from at least three independent experiments. Data are represented as mean ± SD. ****p < 0.0001, *** P < 0.001, **P < 0.01, *P < 0.05, ns: no significant difference. Source data are provided as a Source Data file.

animals (population) after several days of culture (Fig. 7a and Supplementary Fig. 7a). Animals with robust food digestion abilities are expected to grow rapidly and exhibit a larger population, enhancing their adaptation in nature.

Firstly, we asked whether specific bacteria signaling promotes animals' adaptation by consuming inedible food. Under simple food-feeding conditions where SS was combined with single bacteria such as *E. coli*, *E. f*, and *B. s* (Fig. 7a), animals were unable to grow and expand

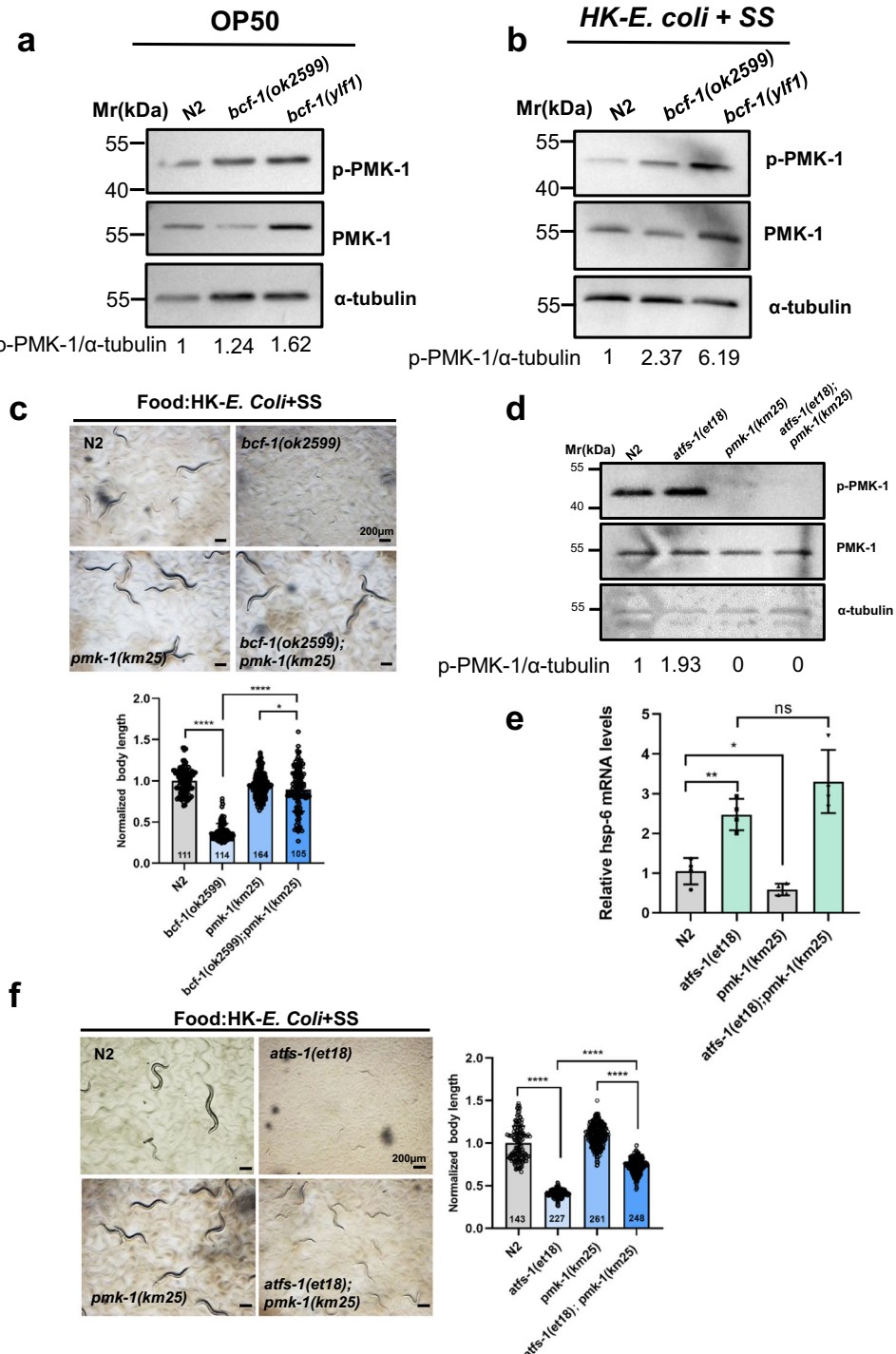

**Fig. 6 | Digestion defects in *bcf-1(ok2599)* and *atfs-1(et18)* mutant with UPR^mt activation depend on PMK-1.** **a** Western blot showing the level of p-PMK-1 in L4 animals grown under OP50 feeding conditions. **b** Western blot showing the level of p-PMK-1 in L1 animals grown under HK- *E. coli* + SS feeding conditions for 24 h. **c** Developmental progression of *bcf-1(ok2599)*, *pmk-1(km25)* and *bcf-1(ok2599);pmk-1(km25)* grown on HK-*E. coli* + SS at 3d at 20°C. *n* = 111 for N2, *n* = 114 for *bcf-1(ok2599)*, *n* = 164 for *pmk-1(km25)*, *n* = 105 for *bcf-1(ok2599);pmk-1(km25)*. Statistical significance was calculated using multiple unpaired *t*-tests (two-tailed). Obtained *p* values were as follows: N2 vs. *bcf-1(ok2599)*; ****p < 0.0001. *bcf-1(ok2599)* vs. *bcf-1(ok2599);pmk-1(km25)*; ****p < 0.0001. *pmk-1(km25)* vs. *bcf-1(ok2599);pmk-1(km25)*; *p* = 0.0491. Scale bar, 200μm. **d** Western blot showing the level of p-PMK-1 in L4 animals grown under OP50 feeding conditions. **e** Results of qRT-PCR analysis showing the expression of *hsp-6* in L4 animals grown under OP50 feeding

conditions. mean ± SD from 3 replicates. Obtained *p* values were as follows: N2 vs. *atfs-1(et18)*; *p* = 0.0015. N2 vs. *pmk-1(km25)*; *p* = 0.0436. *atfs-1(et18)* vs. *atfs-1(et18); pmk-1(km25)*; *p* = 0.1121. **f** Developmental progression of *atfs-1(et18)*, *pmk-1(km25)* and *atfs-1(et18);pmk-1(km25)* grown on HK-*E. coli* + SS at 4d at 20°C. *n* = 143 for N2, *n* = 227 for *atfs-1(et18)*, *n* = 261 for *pmk-1(km25)*, *n* = 248 for *atfs-1(et18);pmk-1(km25)*. Statistical significance was calculated using multiple unpaired *t*-tests (two-tailed). Obtained *p* values were as follows: N2 vs. *atfs-1(et18)*; ****p < 0.0001. *atfs-1(et18)* vs. *atfs-1(et18);pmk-1(km25)*; ****p < 0.0001. *pmk-1(km25)* vs. *atfs-1(et18);pmk-1(km25)*; ****p < 0.0001. Scale bar, 200 μm. For all panels, n= number of animals which were scored from at least three independent experiments. Data are represented as mean ± SD. ****p < 0.0001, *** *P* < 0.001, **P* < 0.01, *P* < 0.05, ns: no significant difference. Source data are provided as a Source Data file.

their population when fed SS alone. The addition of *E. f*, which does not efficiently activate animals to digest SS (Supplementary Fig. 1b), did not result in an increased population. Conversely, the addition of *E. coli* or *B. s*, which efficiently activate food digestion through PGN, led to an enhanced population due to improved food digestion (Fig. 7a, b). Introducing more complex food conditions (Supplementary Fig. 7a) by mixing SS with two or three bacteria resulted in the activation of animals to digest SS and subsequent population growth upon the addition of *B. s* and *E. f*. However, the addition of high-quality food *E. coli* (OP50) had the opposite effect in increasing the animal population under OP50 + SS + *B.s* + *E.f* feeding conditions (Supplementary Fig. 7b). These findings suggest that specific bacteria such as *E. coli* or *B. s* promote animals to digest a wide range of food sources as part of their adaptive strategy.

Subsequently, we investigated whether the bacterial PGN binding protein, BCF-1, contributes to the adaptation of bacteria-eating worms by facilitating the consumption of complex food. Under complex feeding conditions (OP50 + SS + *B.s* + *E.f*), we observed a decrease in the total number of animals in the *bcf-1* mutant, indicating that the interaction between PGN and BCF1 acts as a "good-food signal" that promotes food uptake and animal adaptation in bacteria-eating worms (Fig. 7c).

Furthermore, we assessed whether PGN has a universal effect on the activation of the digestive system by testing other wild-type nematodes isolated from natural environments. Our results showed that wild type ED3077 and JU2513 were able to grow normally on SS + PGN (Fig. 7d).

In conclusion, the findings above suggest that the PGN-BCF1 signaling system plays a key role in enabling bacteria-eating worms to consume a diverse range of food sources, thereby enhancing their adaptation to their natural environment.

## Discussion

The succession of bacterial-feeding nematode species in natural ecosystems is crucial for ecosystem functioning[1]. Therefore, the ability to digest a diverse range of foods is essential for survival. However, the signals from the bacterial food that trigger host food digestion are still unclear. In this study, we re-evaluated our previous finding that low-quality food (HK-*E. coli*) promotes animals to digest inedible food (SS)[12], which may increase the ability of animals to consume a greater variety of food. This fascinating result may underlie a mechanism for the increased fitness of nematodes in nature.

Using the established digestion system and screening bacterial mutants, we identified that bacterial peptidoglycan stimulates animals to digest an inedible food, SS. On the host side, we discovered that intestinal glycosylated protein, BCF-1, interacts with PGN to promote food digestion by inhibiting UPR^mt through neuropeptide signaling. Activation of UPR^mt was observed in animals fed mutant bacteria lacking PGN or by *bcf-1* mutation, both of which exhibited impaired food digestion. Furthermore, we found that constitutive activation of UPR^mt suppresses food digestion by inducing PMK-1, establishing a link between the UPR^mt and the digestive process. Additionally, we also found that BCF-1 inhibits UPR^mt by transmitting signals to neurons to regulate the release of neuropeptide NLP-3, suggesting an involvement of the gut-nerve signaling axis in food digestion. Overall, our study uncovered an intriguing mechanism in bacterial-feeding nematodes for surviving in nature by digesting more complex food through the sensing of bacterial PGN via the intestinal glycosylated protein (BCF-1) to inhibit UPR^mt in the host (Fig. 7e). This study also suggests the beneficial roles of bacterial PGN in activating the food digestive system, as the abundant amount of commensal gut bacteria which contain the PGN in their cell wall in the animal kingdom.

### Conserved function of PGN in promoting animals' growth
Bacterial-feeding nematodes play a crucial role in ecosystems, agriculture, and human health worldwide[22]. They contribute significantly

to decomposition and nutrient cycling in ecological and agricultural systems[2]. The ability of nematodes to consume and digest a wide variety of foods and bacteria is essential for their physiological well-being and survival in natural environments The microbes as food for nematodes are composed of both gram-positive (G+) and gram-negative (G-) bacteria. The cell walls of bacteria contain many components that are recognized by the animals for regulating innate immune responses, such as lipopolysaccharide[23–25], ADP-heptose[26], and peptidoglycan (PGN)[27,28]. PGN, a core element of all bacterial cell walls, has been identified as a key cue for promoting animal growth in mice, as demonstrated by Schwarzer et al.[29].

The study by Schwarzer et al.[29] revealed that PGN from Lp^WJL interacts with the pattern recognition receptor NOD2 in the intestinal epithelium, leading to increased postnatal growth despite nutritional limitations. However, the strain Lp^NIZO2877 did not have the same growth-promoting effect, indicating strain-specific effects of PGN on growth. Evolutionarily, bacterial-feeding nematodes have likely developed mechanisms to sense bacterial PGN as part of their food source, enabling them to thrive in their natural habitats. Building on this concept, our research in the nematode *Caenorhabditis elegans* found that PGN from *E. coli* and *B.s* can enhance animal digestion of normally inedible food, while PGN from *E. faecalis* does not have the same effect. This suggests that the influence of bacterial PGN on growth may vary depending on the bacterial strain.

Recent studies have shown that bacterial PGN muropeptides play a beneficial role in mitochondrial homeostasis and animal growth by acting as ATP synthase agonizts in *C. elegans*[15]. Additionally, components of peptidoglycan, such as MurNAc-L-Ala and MurNAc, have been found to enhance pathogen tolerance in *C. elegans*[30], further highlighting the beneficial effects of PGN in animals. These findings support our hypothesis of co-evolution between microbial PGN and nematodes for adaptation in diverse natural environments rich in food sources and pathogens.

### PGN-BCF1 interaction acts as a specific "good-food signal" that promotes food uptake and animal adaptation in bacteria-eating worms
In hosts, we have demonstrated that the conserved glycosylated protein BCF-1 in nematodes (including *C. remanei, C. nigoni, C. briggsae, C. bovis, C. auriculariae*, and *C. elegans*) interacts with PGN to enhance food digestion by inhibiting the mitochondrial unfolded protein response (UPR^mt). The UPR^mt is a conserved transcriptional response activated by various forms of mitochondrial dysfunction. This mechanism enables BCF-1 in nematodes to sense the unique PGN of bacteria and activate food digestion by regulating the UPR^mt in the host. This specific mechanism of food digestion is exclusive to bacterial-feeding nematodes, as the PGN-binding protein BCF-1 lacks homology in other species, including humans. It represents an efficient survival strategy for bacterial-feeding nematodes, as they rely on bacteria as their primary source of food. In our previous study, we discovered that BCF-1 binds to *E. coli* through its fimbriae, promoting *E. coli* colonization[18]. It is conceivable that BCF-1 facilitates initial *E. coli* colonization, leading to the production of PGN by *E. coli*. Subsequently, PGN interacts with BCF-1 to trigger food digestion. Therefore, the beneficial role of PGN in nematode food digestion through BCF-1 may serve as a causal mechanism for the co-evolution between microbes and nematodes in adapting to a diverse range of food sources.

### Evolutionarily conserved function of PGN in promoting digestion
In this study, we demonstrated that bacterial PGN activates *C. elegans* to digest inedible food *Staphylococcus saprophyticus* through suppress UPR^mt. Does PGN have roles in promoting digestion in mammals? Recently, researchers in Min Han's lab at the University of Colorado Boulder found that bacterial muropeptides promote OXPHOS and

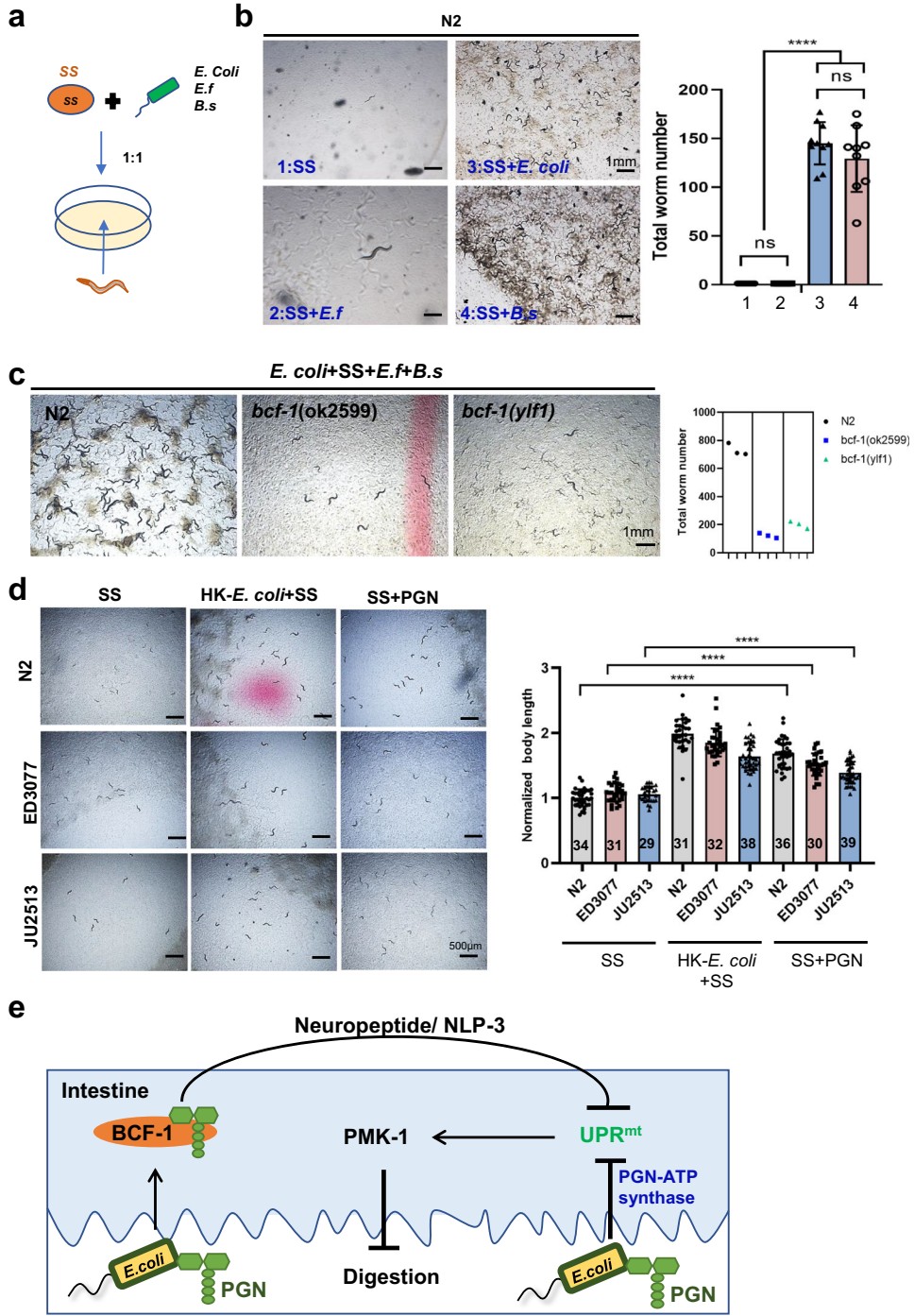

**Fig. 7 | The PGN-BCF1 signaling system enhances animal adaptation. a, b** Adaptation analysis in animals feeding mixed bacteria to mimic nature environment. A mixture of SS with *E. coli, Enterococcus faecalis f*), and *Bacillus subtilis* (*B.s*) at a 1:1 ratio was prepared as food source (**a**). single wild-type N2 animal was then placed on the food. After 9 days of culture, the number of animals on the plate was counted (**b**). In panel b, mean ± SD from 10 replicates. Obtained *p* values were as follows:SS + OP50 vs. SS + B.s; *p* = 0.3345. SS vs. SS + OP50 and SS + B.s; ****p* < 0.0001, respectively. **c** A mixture of *E. coli, Staphylococcus saprophyticus* (SS), *Enterococcus faecalis* (E.*f*), and *Bacillus subtilis* (*B.s*) at a 1:1:1:1 ratio was used to mimic the natural food environment. A single wild-type N2 or *bcf-1* mutant animal was seeded on the food. After 10 days of culture, the number of animals on the plate was scored. **d** Developmental progression of ED3077 and JU2513 grown on SS + PGN at 20°C for 3d. *n* = 34, 31 and 29 for N2, ED3077 and JU2513 on SS. *n* = 31, 32 and 38 for N2, ED3077 and JU2513 on HK-*E. coli* + SS. *N* = 36, 30 and 39 for N2, ED3077 and JU2513 on SS + PGN. Statistical significance was calculated using multiple

unpaired *t*-tests (two-tailed). Obtained *p* values were as follows: N2 on SS vs. SS + PGN; ED3077 on SS vs. SS + PGN; JU2513 on SS vs. SS + PGN; ****p* < 0.0001, respectively. **e** The model illustrates that the PGN-BCF-1 interaction acts as a "good-food signal" to promote food digestion and animal growth in bacteria-eating worms. The conserved glycosylated protein BCF-1 in nematodes interacts with bacterial PGN to inhibit UPR[mt] through neuropeptides (NLP-3), thereby enhancing the ability to digest inedible food (SS). PGN inhibits UPR[mt] through interacting with BCF-1 or ATP synthase[15]. Activation of UPR[mt] inhibits food digestion through inducing PMK-1. This mechanism reveals an intriguing adaptation strategy in animals to survive in nature by increasing their capacity to digest a variety of foods through the detection of the unique bacterial cell wall component PGN. For all panels, *n* = number of animals that were scored from at least three independent experiments. Data are represented as mean ± SD. ****p* < 0.0001, *** *P* < 0.001, ***P* < 0.01, **P* < 0.05, ns: no significant difference. Source data are provided as a Source Data file.

suppress mitochondrial stress in normal and human mitochondrial disease models[31], suggesting that *E. coli* peptidoglycan (PGN) also has evolutionarily conserved role in suppressing the UPR$^{mt}$. They also examined the physiogy function of *E. coli* muropeptides derived from PGN in the intestines of mice[31]. Initially, they explored the impact of muropeptides in the small intestine of mice by depleting the intestinal microbiota using an antibiotic cocktail (AIMD). This depletion resulted in germ-free-like characteristics, such as an enlarged cecum and reduced spleen size in both male and female mice. Given that the small intestine serves as the primary site for nutrient sensing and absorption, the researchers monitored food consumption and stool output in AIMD-treated mice to evaluate the effect of muropeptides on nutrient absorption. While neither AIMD nor PGN feeding induced significant changes in food consumption within 24 h, AIMD-treated mice showed increased stool production during this period, indicating compromised digestion and nutrient uptake due to microbiota depletion. Notably, oral administration of PGN partially but significantly reduced the increase in stool output in male mice, suggesting that PGN muropeptides enhance nutrient absorption. Furthermore, AIMD treatment resulted in reduced weight gain in mice, which was partly offset by oral administration of PGN in male mice. Taken together, these findings suggest that muropeptides promote nutrient absorption in the small intestine of mice. Therefore, PGN may have a conserved function in enhancing food digestion in both *C. elegans* and mice. If its role in digestion in mammals can be confirmed in the future, it could potentially be developed as a therapeutic agent for treating digestive disorders.

## Why does UPR$^{mt}$ activation inhibit food digestion?

Our study has shown that UPR$^{mt}$ activation inhibits food digestion and utilization in animals. The UPR$^{mt}$ is a transcriptional response that is activated in response to mitochondrial dysfunction and is regulated through communication between the mitochondria and nucleus[32]. Its activation is vital in repairing and restoring mitochondrial function to maintain cellular homeostasis.

Physiologically speaking, the activation of UPR$^{mt}$ leads to a shutdown of food digestion in animals, with the aim of reducing protein translation and cellular metabolism. This response is reminiscent of a state of starvation and promotes cellular stress alleviation and recovery. In our study, we investigated this unique biological function of UPR$^{mt}$ activation in regulating food digestion and behavior. Additionally, our research provides new insights into the physiological mechanisms that regulate the food digestive system and feeding behavior, which may have important implications for our understanding of dietary adaptation in animals.

Collectively, this study reveals that unique bacterial PGN was sensed by conserved glycosylated protein (BCF-1) for food digestive system activation in nematodes, which is important for animals in nature enabling them to digest more diverse food, increasing their fitness. In addition, by sensing PGN the food digestive mechanism reported here could have a significant influence on our understanding of the mechanism of how animals adapt to the abundance of food diversity by regulating its digestive system and feeding behaviors.

## Methods

### *C. elegans* strains and maintenance

Nematode stocks were maintained on nematode growth medium (NGM) plates seeded with bacteria (*E. coli* OP50) at 20 °C.

The following strains/alleles were obtained from the Caenorhabditis Genetics Center (CGC) or as indicated:

N2 Bristol (wild-type control strain);
ED3077;
JU2513;
RB1971[*F57F4.4(ok2599)*];
SJ4100: zcIs13 [hsp-6p::GFP + lin-15(+)];
MGH171: sid-1(qt9) V, alxIs9 [vha-6p::sid-1::SL2::GFP]

SJ4197: zcIs39 [dve-1p::dve-1::GFP].
QC118: atfs-1(et18)
YNU30: bcf-1(ylf1);
YNU323: bcf-1(ok2559);pmk-1(km25) double mutant was constructed by crossing RB1971[*F57F4.4(ok2599)*] with KU25[*pmk-1(km25)*].
YNU347: atfs-1(et18);pmk-1(km25) double mutant was constructed by crossing QC118[*atfs-1(et18)*] with KU25[*pmk-1(km25)*].
PHX4067: syb4067[bcf-1p::bcf-1::3xflag::gfp(knock-in)];
YNU380: bcf-1(ok2599), hsp-6p::GFP was constructed by crossing RB1971[*F57F4.4(ok2599)*] with SJ4100: zcIs13 [hsp-6p::GFP + lin-15(+)].

### Bacterial strains

*E. coli*-OP50(from Caenorhabditis Genetics Center (CGC)), *E. coli*-K12 (BW25113), *E. coli*-K12 mutant(from Dharmacon), and *Staphylococcus saprophyticus*(from ATCC) were cultured at 37 °C in LB medium. A standard overnight cultured bacteria was then spread onto each Nematode growth media (NGM) plate.

### Heat-killed *E. Coli* preparation

Heat-killed *E. coli* was prepared by an established protocol[33]. A standard overnight bacterial culture was concentrated to 1/10 vol and was then heat-killed at 80 °C for 2h.

### Preparation of *S. saprophyticus* (ATCC 15305) and HK-*E. coli* + *S. saprophyticus* (SS)

*S. saprophyticus* and HK-*E. coli* + SS were prepared by following our established protocol[12]. For SS preparation, a standard overnight culture of SS (37 °C in LB broth) was diluted into fresh LB broth (1:100 ratio). SS was then spread onto each NGM plate when the diluted bacteria grew to OD600 = 0.5. For HK-*E. coli* + SS preparation, HK-*E. coli* (50ul) and SS (50ul) were mixed at a 1:1 ratio, then 100ul of the mixture was spread onto NGM plates.

About 100–200 synchronized L1 worms were then seeded onto the indicated plates (SS, or HK-*E. coli* + SS) and cultured at 20 °C for growth phenotype obversion.

### *E. coli* Keio collection screen

*E. coli* mutants are from the Keio *E. coli* single mutant collection[34]. Mutant bacteria strains, as well as the wild-type control strain BW25113, were cultured overnight in LB medium with 50 ug/ml kanamycin in 96-well plates at 37 °C. Overnight cultured bacteria were then heat-killed following our established protocol[33]. *S.saprophyticus* was also prepared as described above. HK-*E. coli* mutants (50ul) and *S.saprophyticus* (50ul) were mixed at a 1:1 ratio and then 100 ul of the mixture was spread onto 3.5 cm NGM plates. About 100–200 synchronized L1 worms were then seeded onto HK-*E. coli* + SS feeding plates and cultured for 3-4 days at 20 °C. The animals' size was then observed.

### RNAi treatment

All feeding RNAi experiments used bacterial clones from the MRC RNAi library[35] or the ORF-RNAi Library[36]. RNAi plates were prepared by adding IPTG with final concentration of 1mM to NGM. Overnight cultured RNAi strains (LB broth containing 100 ug/ml ampicillin) and the control strain (HT115 strain with empty L4440 vector) were seeded into RNAi feeding plates and cultured at room temperature for 2 days before use.

### RNAi screen

Developmental screening on HK-*E. coli* + SS: Synchronized L1 worms were treated by feeding candidate RNAi for the second generation and grew to adult. They were then bleached and allowed to hatch in M9 buffer for 18hr. The synchronized L1 worms were seeded on the feeding plate (HK-*E. coli* + SS). The worm development phenotype (body length) was measured after culturing 3-4 days at 20 °C.

Neuropeptide screening: Synchronized L1 hsp-6p::GFP transgenic animals were treated by feeding neuropeptide candidate RNAi for the

second generation and grew to adult. They were then bleached and allowed to hatch in M9 buffer for 18hr. The synchronized L1 nematodes were then fed *bcf-1* RNAi and the *hsp-6p*::GFP fluorescence level was observed as adults grew to adults.

## Food avoidance behavior assay
Food avoidance assay was performed according to published method[15]. 20ul of overnight cultured *E. coli-OP50* was seeded on the center of 6cm NGM plates. About 100 synchronized L1 animals were seeded onto the bacterial lawns and cultured at 20 °C for 48 h. The avoidance index was determined by $N_{off}/_{on}/N_{total}$.

## Observation of intestinal bacterial accumulation
L1-stage nematodes were placed on SS, SS + PGN or HK-*E.coli* + SS at 20 °C for 4 days and examined under a 100× microscope. If animals are unable to digest bacteria, bacteria will accumulate in the lumen, resulting in a distended lumen. The width of the intestinal lumen was measured in animals fed different types of food.

## PGN extraction, enzyme treatment and supplementation assays
PGN was extracted according to an established method[15]. In brief, bacterial pellets from a standard overnight culture of *E. Coli* OP50 (37°C in LB broth) were resuspended in 1/10 vol 1M NaCl solution and boiled at 100 °C in a heating block for 30min. After washing 3 times by ddH$_2$O, the insoluble cell fragments were crushed by ultrasound for 1h, followed by stepwise digestion using DNAse (50ug/ml), RNAse (60ug/ml), and trypsin (50ug/ml) at 37°C for 1h, respectively. After collecting bacterial pellets and washing them with ddH$_2$O three times, they were heated at 100 °C for 5 min to inactivate the enzymes. Finally, the pellets collected by centrifugation at 12,000g were re-suspended with HEPES buffer (20mM, pH=7.5) and stored at -20°C.

For enzymatic treatment of isolated PGN, lysozyme (500ug/ml), NagZ (purified from BL21 *E. coli*,500ug/ml), AmiD (purified from BL21 *E. coli*, 500ug/ml), or Protease K (500ug/ml) were added to isolated PGN and incubated at 37°C for 24h. The reaction was terminated by adding trypsin (500ug/ml) at 37°C for 2h. Finally, the above samples were heated at 100°C for 5 min to inactivate the enzymes.

For supplementation assays, indicated bacteria (SS, *ΔycbB*) and PGN solution mixtures (1:1 ratio) were seeded on NGM plates. The synchronized L1 worms were then seeded on the indicated plate (SS, SS + PGN, or SS+enzyme treated PGN; *ΔycbB*, *ΔycbB* + PGN).

## Protein expression and purification
The expression of recombinant proteins (NagZ and Amid) was performed by established protocols[15]. Briefly, the NagZ and AmiD genes were amplified by PCR from *E. coli-K12* (BW25113) genomic DNA. Then the target genes were inserted into pET28a vector by homologous recombination. These constructs were transformed into *E. coli BL21(DE3)*. Overnight cultured BL21 bacteria were diluted into fresh LB broth (1/100 vol ratio) and cultured to OD600 ≈ 0.6. Then, the expression of recombinant proteins was induced by 1 mM IPTG at 20 °C for 19 h. Bacterial pellets were collected by centrifugation at 5000g for 10min at 4°C, and the pellets were lysed in buffer (25 mM Tris-HCl pH = 8.0, 150Mm NaCl, 10% Glycerol, 0.1% NP40). The suspension was ultrasonically broken (25%power, followed by centrifugation at 12,000g for 30min at 4 °C. The supernatants were incubated with Ni-NTA agarose beads (TianGen 30210) and rinsed 3 times with a washing buffer (25mM Tris-HCl pH = 8.0, 150mM NaCl, 10% Glycerol, 0.1% NP40, 30mM Imidazole). Finally, the Elution buffer (25mM Tris-HCl pH = 8.0, 150mM NaCl, 10% Glycerol, 0.1% NP40, 500mM Imidazole) was used to eluted the recombinant protein. Amicon Ultra centrifugal filters (10kD) were used to concentrate the proteins and exchange buffer (25 mM HEPES pH 7.5, 250 mM NaCl, and 1 mM DTT).These proteins were stored at -80 °C.

## Worm total protein extraction
Total proteins were extracted according to our established method[18]. Worms were lysed by freeze-thawing three times in protein lysis buffer containing PMSF (1mg/ml) in liquid nitrogen. The worms were then ground in a tissue grinding tube. The resultant liquid was then centrifuged at 12,000g for 30min at 4 °C, and the supernatant was taken as the total protein of worms. The protein concentration was measured by using the Pierce BCA protein assay kit (ThermoFisher, 23227).

## BCF-1 expression assay with PGN induction
Synchronized L1 animals (*bcf-1p::bcf-1::gfp::flag*) were seeded onto SS or SS + PGN feeding plates and cultured at 20 °C 4 days before observing the fluorescence.

## PGN-protein binding assay
The PGN-protein binding assay was carried out as described[37]. Briefly, total protein (1mg) of *C. elegans* was incubated with PGN at 4°C for 4h. At the end of incubation, the PGN was pelleted, then the pellet was washed 3 times with washing buffer (100 mM NaCl, 50 mM Tris-HCl pH = 7.5). Proteins associated with the pellet and the incubated supernatant were detected by Western blot with an anti-flag antibody.

## Analysis of the fluorescence intensity in worms
For fluorescence imaging (*hsp-6p::gfp*), worms were anesthetized with 25 mM levamisole, and photographs were taken using an Olympus MVX10 dissecting microscope with a DP80 camera. To quantify fluorescent intensity, the entire intestine regions were outlined and quantified using ImageJ software. The fluorescence intensity was then normalized to the body area.

For quantifying BCF-1::GFP, the worms were mounted on 2% agarose pads with 25 mM levamisole and imaged using an Olympus MVX10 microscope with a DP80 camera.

## Real-time PCR
**Sample preparation.** L1-stage nematodes (N2, *pmk-1(km25), atfs-1(et18), atfs-1(et18); pmk-1(km25)*) were obtained by bleach and placed on OP50 and cultured until L4 stage.

**RNA isolation.** The prepared samples were quickly collected and 500 ul Trizol (Invitrogen) was added. Liquid nitrogen three freeze three melt. Added chloroform (200ul) to extract RNA. Followed by isopropanol precipitation (400 ul) of the aqueous phase, and a single wash of the resulting pellet with 75% ethanol. Finally, RNA pellets were dried in a tissue culture hood and re-suspended in RNase-free water.

**cDNA synthesis.** The cDNA was reversed according to the reverse transcription kit (Takara) protocols.

**qPCR reaction.** qPCR was performed by using PowerUp™ SYBR™ Green (Thermo Fisher A25742) on a real-time PCR machine (ABI-QuantStudioI). Relative gene expression levels were calculated using the 2-ΔΔCt method.

## Primers
*tba-1* (Forward):TCAACACTGCCATCGCCGCC,
   *tba-1* (Reverse):TCCAAGCGAGACCAGGCTTCAG;
   *hsp-6* (Forward):CAAACTCCTGTGTCAGTATCATGGAAGG,
   *hsp-6* (Reverse):GCTGGCTTTGACAATCTTGTATGGAACG

## Nematode adaptation assay
Overnight-cultured four kinds of bacteria (*E. Coli, SS, E.f* and *B.s*) were mixed at the 1:1:1:1 ratio. The mixed bacteria were spread onto NGM plate. Then, seed one L1 staged animals and measure the number of animals on the plate after being grown at 20 °C for 10 days.

## Protein phosphorylation level detection by western blot

Synchronized L1 nematodes grow on OP50 and HK-*E. coli* + SS respectively for indicated time (see Fig. 6). Then collect the animals and extract the total protein for detecting p-PMK-1 level by standard Western blot methods and probed with following antibodies: Monoclonal anti-a-tubulin antibody (dilution = 1:10000, Sigma, T5168), anti-p-p38 (dilution = 1:5000; Cell Signaling, 4511S).

## Statistics and reproducibility

ImageJ software was used for quantifying the fluorescence intensity of various reporters or analysis of the developmental body length of nematodes. The fluorescence intensity was normalized to the body area. Statistical analysis was performed with Prism 8 (Graphapd Software Inc.). For statistics, significance was determined at $P < 0.05$. To ensure reproducibility, different alleles of each gene were used for phenotypic analyses, with data typically generated from three biological replicates, each containing at least three technical replicates. All data are expressed as mean ± SD. The Investigators were not blinded to allocation during experiments and outcome assessment.

## Reporting summary

Further information on research design is available in the Nature Portfolio Reporting Summary linked to this article.

## Data availability

All data in the main Manuscript and Supplementary information are listed in the Source data file. All reagents and strains generated by this study are available through a request to the corresponding author with a completed Material Transfer Agreement. Source data are provided in this paper.

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

## Acknowledgements

We thank the Caenorhabditis Genetics Center (CGC) (funded by NIH P40OD010440) for strains; and Dr. Zhao Shan for suggestions; Dr. Leonard Krall and Zhao Shan for editing help; Our lab members (Qian Li, Yating Liu and Yongjuan He) for sharing strains. This work was supported by the Ministry of Science and Technology of the People's Republic of China (2019YFA0803100, 2019YFA0802100), the National Natural Science Foundation of China (32170794), Yunnan Provincial Science and Technology Project at Southwest United Graduate School (202302AP370005), Yunnan Applied Basic Research Projects (202201AT070196).

## Author contributions

F.H designed research, performed all most experiments, analyzed data and revise the paper. H.L performed RNAi screen (Supplemental Fig. 2) and some UPR$^{mt}$ experiments (Fig. 3b, Fig. 4c–e, Fig. 5a). B.Q. designed research, supervised this study, and wrote the paper with inputs from F. H.

## Competing interests

The authors declare no competing interests.
