## [Peer Review File · Nature Communications]

REVIEWER COMMENTS

Reviewer #1 (Remarks to the Author):

General comments:

In this paper, Hao et al., present a novel mechanism underlying the bacteria-host interaction that promotes food digestion. In a paper published last year, the authors' lab presented data to indicate that the p38 innate immunity pathway mediates a bacteria-host interaction that activates the food digestive system in worms. In this new manuscript, the group made a potentially very significant advance in understanding the food signal. The key novel findings are: (1) bacteria peptidoglycan (PGN) acts as the signal that stimulates food uptake and animal growth; (2) a glycosylated protein BCF-1 is the host factor that likely acts as the receptor for PGN in the intestine for the function; (3) the PGN-BCF-1 activity on digestion/growth is mediated by unfolded protein responses in mitochondria (UPRmt); and (4) a neuropeptide NLP-3 may act upstream of UPRmt to mediate or facilitate the signal. The authors also investigated the connection between this pathway and the previously identified function of PMK-1/p38, showing that p38 acts downstream of BCF-1. Some of these indications were supported by solid data, but some conclusions are either not fully supported or need modification in data interpretation.

This reviewer found the problem addressed by this study to be of high interest to the relevant fields. The findings, if substantiated, are novel and significant. I would support the permission for a major revision.

Although it did not significantly affect my ability to understand the work, the current manuscript has many obvious grammatical errors and writing mistakes. The professional editing service the authors used did not do the needed job. The authors need to put serious effort into improving the writing, perhaps using a different language editor.

Specific comments

Major issues:

1. Starting from Fig.1 and throughout the paper. The authors used animal growth assays to represent digestive activity. While digestion affects growth, growth defects from knocking out genes or feeding bad food could be caused by other reasons. It is understandable that the authors did this because the 2022 paper had shown that food was not consumed in worms fed SS bacteria and the expression of three glucosidases were regulated by the quality of food. However, this paper analyzes the role of many new factors that could have pleiotropic roles, so the connection to digestion, not growth, needs to be established by experiments at least in some critical conditions. For example, UPRmt may not impact growth only through inhibiting digestion. Also, in WormBase, *bcf-1(-)* is reported to cause

multiple severe phenotypes so it may also have roles beyond digestion. The same question may be asked for PGN.

2. Fig. 2 and Fig. 7. Have the authors tried to use PGN isolated from other bacteria species such as SS, E.F and B.S? It seems to be important for this paper to know if the signal only exists in high quality food like E. coli, as the authors proposed an adaptation hypothesis.

3. In Fig. 4, the data to support that *bcf-1(-)* causes UPRmt and that UPRmt likely acts downstream of BCF-1 for the growth function look solid. In Fig. 4a, feeding *ycbB(-)* bacteria to *bcf-1(-)* worms did not cause an additive effect on *hsp-6* level, which is consistent with a linear relationship between *ycbB* (PGN) and BCF-1. However, the data in Fig.4a indicated that feeding *ycbB(-)* bacteria had a much stronger impact on *hsp6* expression than did *bcf-1(-)*, suggesting that only some percentage of the effect of PGN on UPRmt goes through BCF-1. This relatively small effect by *bcf-1(-)* is still very significant but would suggest the presence of a BCF-1-independent effect of PGN on UPRmt. This point, if agreeable by the authors, should be made in the text below line 188.

On line 191, the authors mentioned that the work in another paper had indicated the role of PGN mucopeptides on UPRmt through acting as an ATP synthase agonist. Perhaps such direct interactions between PGN and mitochondrial proteins represent at least some of the BCF-1-independent impact of PGN on UPRmt. The authors should discuss this issue to help the reader to understand the difference and relationship between these two pathways.

Besides the UPRmt assay, the authors might also want to have an animal growth assay to compare the effect of feeding *ycbB(-)* bacteria (wild-type worms) and mutating *bcf-1* [feeding K12 and *ycbB(-)* bacteria]. Comparing them under the same condition as in Fig. 4A or Fig. 1D would be helpful regarding the above issue (also mentioned below).

4. Fig. 6B-D. The data for PMK-1 acting downstream of BCF-1 look solid, but the authors did not provide a strong support for UPRmt acting downstream or upstream of PMK-1. In Fig. 6D, while the growth of the *atfs-1(et18)*(increased UPRmt) mutant was improved by mutating *pmk-1*, the growth of the double mutant was lower than that of *pmk-1(-)* alone and the difference in the latter is much greater. From the data, we could also argue that *atfs-1(gf)* suppressed the beneficial effect of *pmk-1(-)* in growth, which could be used to suggest that PMK1 acts upstream of UPR. Because UPRmt and PMK-1 play roles in, or impact, many different signaling pathways, it is not reasonable to make too much out of the animal growth assay that could be affected by multiple pathways. If the authors want to suggest that PMK-1 acts downstream or upstream of UPRmt for digestion and growth, then more molecular markers would be needed for the epistasis analysis.

It is noticeable that the authors did not place PMK-1 in the model in Fig. 7C, as they probably noticed the issue. Since the role of PMK-1 was a key finding in their 2022 paper, it is good that the relationship with PMK-1 was investigated. Besides potentially doing additional tests to address the

issue mentioned above, the authors may make significant modifications to describe the result related to Fig. 6D (line 266-273) and perhaps also discuss the potential relationship in the Discussion.

5. The data in Fig. 1A and Fig. 7A seem to be the basis for the authors to propose the interesting hypothesis that PGN-BCF1 signaling is a system that facilitates adaptation of bacteria-eating worms to increase the ability to consume a wide range of food in their natural environment. However, the rationale for the hypothesis does not seem to be sufficient, or it is at least not well explained. In Fig. 1A, it is okay to call HK-E. coli low-quality food, based on its ability to support growth, but because it is artificially made (heating) from "high quality food" E. coli, the result might not have much power to model what is found in nature. As for the experiment in Fig. 7A, because only one condition was tested using only a growth assay, the result may raise multiple questions. Are we assuming only E. coli can produce the signal? Are we assuming BCF-1 does not interact with anything from other bacterial species? To increase the significance of the test, the authors may want to design additional tests with different combinations. A control with 100% E. coli and 0% E. coli would need to be part of it. As mentioned earlier, whether the other bacterial species (E.f and B.S) can produce functional PGN is not clear. These data seem to be important for forming the hypothesis.

At this point, the data seem to support a model that the PGN-BCF-1 interaction acts as a "good-food signal" to promote food uptake and animal growth in bacteria-eating worms. The authors might stay with such a notion, which is already novel and significant, in the abstract and the text. They might discuss additional speculative ideas in the Discussion.

6. The model figure in Fig. 7C needs significant modification. (1) The authors need to have a better figure legend to describe/explain the arrows and bars, particularly since the model is not specifically described in the Discussion where the figure is cited. If based on the data in the lab's other papers, citations may be made. (2) If the authors agree with the point in 3 above, PGN should have a direct inhibitory bar to UPRmt, which may be mentioned/explained in the legend. (3) It is not clear why the authors have a big blue arrow pointing directly from BCF1 to inedible food. The data in Fig. 4 and 5 seem to support UPRmt acting downstream of BCF1 and it is not clear what data supports a UPRmt-independent function of BCF1. Perhaps the authors were thinking about the epistasis data involving PMK1 that seem to indicate that PMK1 acts downstream of BCF1 but not necessarily downstream of UPRmt (mentioned under 4 above). However, there is also no clear evidence against the idea that PMK1 is acting upstream of UPRmt (additional tests could clarify this). Without seeing a UPRmt-independent BCF1 activity, this arrow seems unnecessary, regardless of the PMK1 issue. The authors may need to re-think these complicated issues to improve the figure. The figure needs to be made to match the proposal with data and their interpretations. (4) Ideally, PMK-1 should be in the model. If not, the authors may need to mention its known relationship with other factors in the legend.

(5) In the model, "inedible food" is confusing. It might be changed to digestion of inedible bacteria. (6) Is the signaling system also important for digesting high quality food? If yes, you may take out inedible food from the figure (use just "digestion" or "digestion of edible and inedible bacteria").

Mentioned as a minor point below, the linear relationship between BCF1 and NLP3 is suggested by the data but might not be concluded. An alternative way to illustrate this relationship would be to have a curved inhibitory T-bar from BCF-1 to UPRmt, and then write “neuropeptides/NLP-3” next to the arrow to indicate participating or required factors, which can be described in the legend. Potentially, PMK-1 could be put in the same spot of the pathway with the explanation in the legend.

Comments on relatively minor issues:

- Fig 1A. on row (d), calling it "low nutrition food" only in this row does not make good sense. The authors should still use "Low Quality Food". Several previous studies have described the micronutrient deficiencies of Heat-killed E. coli. The authors may want to add a sentence in the Results or Introduction to describe these defects to define this food.

- Fig. 1C is confusing. A better diagram is needed to illustrate the screen.

- Fig. 2B and 2C. As mentioned above under #3, it would be good to have a test on worms fed K12 E. coli as in Figure 4a or Fig. 1D so that ycbB(-) feeding could be added as a control (the test in Fig.1D). In this way the extent of the effects under the two conditions can be compared, which may help to address the issue mentioned under 3 above.

- Fig. 3. This lab found BCF-1 as a protein binding to E. coli in a previous publication, and also mentioned that BCF-1 was separately identified as a PGN-binding protein in another paper. Since E. coli binding is involved in bacterial colonization in the worm (He et al paper), such an interaction seems to involve the large/intact PGN molecule on the bacterial surface. Therefore, it is not clear if the bacterial binding discussed in the other publications is the same as the binding shown in this paper. It would be good for the authors to clarify this. By the way, does the binding activity in either case depends on lipoprotein on bacterial membrane? Lipoprotein could be eliminated by certain site-specific proteases.

- Fig. 3D. PGN addition caused an increased level of BCF-1:GFP. PGN seems to promote the expression of bcf-1. Is the regulation at the transcriptional level? It is not common that the presence of signal also regulates the expression of the receptor, which could be interesting if confirmed. What is the potential theory on this? The authors may want to comment on it.

- Fig. 5E, the test was done using only the *bcf-1(-)* mutant worms, which is effective to show the rescue of *bcf-1(-)* defect in growth. However, it would be helpful to have the wild-type worms as a control so that the extent of the rescue is seen. Is the ~30-50% increase in body length sufficient to suppress most or only part of the defect? This would not take away the suppression role but may help to understand the *bcf-1* roles and the pathway.

- Similarly, for Fig 6C and D, having a wild-type worm control would help to learn the extent of suppression by *pmk-1(-)*, that would help to view its function in the pathway.

- In Fig. 5, the data supporting the roles of *nlp-3*, *egl-3*, and *elg-12* on UPRmt are solid. The suppression data by RNAi of these genes in the *bcf-1(-)* mutants provided further support for UPRmt acting downstream of BCF-1. It is also reasonable to state, as the authors did, that the suppression data suggested that *nlp-3* acts downstream of BCF-1, even though the extent of suppression is not clear based on the data in Fig. 5D&E (lack of positive control such as *bcf1(wt)* animals). However, there is no evidence that the 1.43-fold expression increase (Fig. S4B) would be the driver of the phenotype. Contributions could also be made by other NLPs. At this point, the safe thing to say is that the *bcf-1(-)* impact on UPRmt at least partially depends on NLP-3. The T-bar in the model from BCF-1 to NLP-3 is a suggestive action.

- Line 244. Figure S5 should be Figure S4.

- Fig. 6A: a non-phosphorylated PMK-1 level for loading control would be better than the tubulin control to evaluate the significance of the change seen in p-PMK-1.

- Sample size. in this study, the sample sizes in some experiments seem quite low. In some cases, the legends indicate that N= the number of animals scored from at least three independent experiments, which would mean even smaller number of worms were scored per condition/experiment. These numbers are low for a worm study. The data and conclusions can only be improved by increasing the sample sizes.

Reviewer #2 (Remarks to the Author):

Review on the manuscript entitled „Bacterial peptidoglycan as a food digestive signal that facilitates adaptation of animals in nature” by Qi and colleagues. The authors used the nematode *Caenorhabditis elegans* as a tractable genetic model feeding on various bacteria (mainly *Escherichia coli*) and other microorganisms (so, consumes a wide range of foods) to demonstrate that bacterial peptidoglycan (PGN) produced by *E. coli* cells is a food signal for the host to digest other, normally inedible food resources (*C. elegans* does not grow on *Staphylococcus saprophyticus* bacteria, but can do when it is also exposed to a low-quality food resource such as heat-killed *E. coli* cells). They showed that the host glycosylated protein BCF-1 (bacterial colonization factor) directly interacts with the bacterial PGN in the gut to promote the release of NLP-3 (neuropeptide-like protein) neuropeptide, which eventually inhibits the mitochondrial UPRmt (unfolded protein response) in the gut cells (a cell non-autonomous mechanism). Constitutively active UPRmt indeed prevents food digestion in an innate immunity protein PMK-1-dependent manner. These are very important results with potent human implications (highlighting the importance of the human microbiome in adaptation and survival). Experiments are well-designed and data are solid. However, it would be important to show that this phenomenon is evolutionarily conserved to higher taxa. To publish the study in *Nat Commun*, I guess it would be important to show that the underlying molecular machinery may also present in higher animal taxa (mammals) despite the fact that some of the key components (e.g. BCF-1 orthologs) are missing from higher organisms. So, the authors should demonstrate the evolutionary conservation of the phenomenon they discovered here or show that it is a nematode-specific trait. Thus, I propose some additional experiments to complete successfully the story.

Major comments:

- 1, The authors should demonstrate that *E. coli* PGN can (or cannot) influence intestinal UPRmt in mammals (humans or mice).
- 2, They should investigate whether a PGN mutant diet inducing UPRmt can result in increased food avoidance behavior (in mice).
- 3, It should be discussed which bacterial taxa contain PGN (does *S. saprophyticus* contain or not, and if yes, why it is ineffective in this respect).
- 4, It is clear that inhibiting the UPRmt leads to food diversification in this organism. However, what is the biology behind this phenomenon, what is the function of UPRmt/innate immune pathway in this response (why induced UPRmt results in increased food avoidance behavior)? How does UPRmt induces the innate immune pmk-1 pathway and what is the role of the pathway in food selection?

Minor points:

- 1, The text, mainly the abstract, should be polished throughout.
- 2, Title: my suggestion: Bacterial peptidoglycan as a food digestive signal that facilitates host adaptation to consuming more diverse food resources.
- 3, Abstract: abbreviations should be shown in detail! BCF-1 (bacterial colonization factor), UPR (unfolded protein response), NLP-3 ..., PMK-1, ...

My suggestion: Food availability and usage is a major adaptive force for the successful survival of animals in nature. However, very little is known about the signal from food to activate the hosts digestive system, which facilitates animals to digest more diverse food in nature.

change to:

Food availability and usage is a major adaptive force for the successful survival of animals in nature. Despite its physiological and medical importance, very little is known about the food signal that activates the host digestive system to facilitate the consumption of more diverse foods in nature.

„We identified that a glycosylated protein BCF-1 in the gut that interacts ...” change to

„We identified that the host’s glycosylated protein ... (BCF-1) in the gut interacts ...”

Text:

4, Fig. 1 (and also in the relevant part of the text): SS should be written „*S. saprophyticus*” or at least “*S. saproph.*”

5, line 131: “the slow growth phenotype of worms fed SS” should be changed to “the slow growth phenotype of worms fed on SS”

6, On some figure panels, *E. Coli* should be changed to *E. coli*

7, line 229: the classical pathway of UPR_{mt} should be changed to the canonical pathway of UPR_{mt}

8, line 263: “(developmental delay) in *bcf-1* mutant was recovered in double mutant (Figure 6B). This should be Figure 6C.

9, Line 267: “by continuously activation of UPR_{mt} in” should be “continuous activation of UPR_{mt}”

10, line 271: digestion in *pmk-1* mutant was also inhibit by – rewrite the sentence

11, line 338: “nematodes could evolute a mechanism” should be “nematodes could evolve a mechanism”

point-by-point responses

REVIEWER COMMENTS

Reviewer #1 (Remarks to the Author):

General comments:

In this paper, Hao et al., present a novel mechanism underlying the bacteria-host interaction that promotes food digestion. In a paper published last year, the authors' lab presented data to indicate that the p38 innate immunity pathway mediates a bacteria-host interaction that activates the food digestive system in worms. In this new manuscript, the group made a potentially very significant advance in understanding the food signal. The key novel findings are: (1) bacteria peptidoglycan (PGN) acts as the signal that stimulates food uptake and animal growth; (2) a glycosylated protein BCF-1 is the host factor that likely acts as the receptor for PGN in the intestine for the function; (3) the PGN-BCF-1 activity on digestion/growth is mediated by unfolded protein responses in mitochondria (UPRmt); and (4) a neuropeptide NLP-3 may act upstream of UPRmt to mediate or facilitate the signal. The authors also investigated the connection between this pathway and the previously identified function of PMK-1/p38, showing that p38 acts downstream of BCF-1. Some of these indications were supported by solid data, but some conclusions are either not fully supported or need modification in data interpretation.

This reviewer found the problem addressed by this study to be of high interest to the relevant fields. The findings, if substantiated, are novel and significant. I would support the permission for a major revision.

Although it did not significantly affect my ability to understand the work, the current manuscript has many obvious grammatical errors and writing mistakes. The professional editing service the authors used did not do the needed job. The authors need to put serious effort into improving the writing, perhaps using a different language editor.

Response: We appreciate the reviewer's kind suggestions and comments, as they have been extremely helpful in improving our study.

Specific comments

Major issues:

1. Starting from Fig.1 and throughout the paper. The authors used animal growth assays to represent digestive activity. While digestion affects growth, growth defects from knocking out genes or feeding bad food could be caused by other reasons. It is understandable that the authors did this because the 2022 paper had shown that food was not consumed in worms fed SS bacteria and the expression of three glucosidases were regulated by the quality of food. However, this paper analyzes the role of many new factors that could have pleiotropic roles, so the connection to digestion, not growth, needs to be established by experiments at least in some critical conditions. For example, UPRmt may not impact growth only through inhibiting digestion. Also, in WormBase, bcf-1(-) is

reported to cause multiple severe phenotypes so it may also have roles beyond digestion. The same question may be asked for PGN.

Response:

Our previous study (Geng et al., 2022) revealed that when *C. elegans* were fed *Staphylococcus saprophyticus* (SS), several observations were made: i) the worms arrested development at early larval stages, ii) SS was not digested but instead accumulated in the gut, and iii) the intestinal lumen of worms fed SS was bloated, suggesting an inability to digest living SS.

1) In this study, we discovered that *E. coli* cell walls peptidoglycan (PGN) activates animals to digest SS, thereby promoting animal growth. To assess the efficiency of food digestion, we measured the width of the intestinal lumen in animals, which becomes bloated when animals consume SS (New data in Fig. 1e). Our findings indicate that the bloated intestine is inhibited in animals supplemented with PGN, suggesting that PGN promotes the digestion of SS.

2) Furthermore, our study also revealed that HK-*E. coli* promotes animals to digest SS, but this process requires the PGN-binding protein, BCF-1. We observed that the intestine was still bloated in *bcf-1* mutant animals fed HK-*E. coli*+SS (New data in Fig. 2d), indicating a decrease in the ability to digest SS in these mutants.

Fig. 1e Microscope image and bar graph show the relative width of intestinal lumen in N2 animals fed SS or SS+PGN food.

Fig. 2d Microscope images and bar graph show the relative width of the intestinal lumen between N2 and *bcf-1* mutants when fed HK-*E. coli* +SS.

2. Fig. 2 and Fig. 7. Have the authors tried to use PGN isolated from other bacteria species such as SS, E.F and B.S? It seems to be important for this paper to know if the signal only exists in high quality food like *E. coli*, as the authors proposed an adaptation hypothesis.

Response:

As per the reviewer's suggestion, we conducted experiments where we extracted PGN from *E. coli*, *Bacillus subtilis* (*B.s*) and *Enterococcus faecalis* (*E.f*). We then added these PGN to the SS food and observed the effects.

Our findings indicate that PGN from *E. coli* and *B.s* are effective in promoting animal growth. However, when we used PGN from *E.f*, the effect of promoting growth was significantly reduced (New data in Supplementary Fig. 1b). We should note that *E.f* is considered a low-quality food that cannot adequately support animal growth, as mentioned in a previous study (Geng et al., 2022). Therefore, these results suggest that specific PGN found in *E. coli* and *B.s* are responsible for activating animals to digest SS food.

3. In Fig. 4, the data to support that *bcf-1(-)* causes UPR_{mt} and that UPR_{mt} likely acts downstream of BCF-1 for the growth function look solid. In Fig. 4a, feeding *ycbB(-)* bacteria to *bcf-1(-)* worms did not cause an additive effect on *hsp-6* level, which is consistent with a linear relationship between *ycbB* (PGN) and BCF-1. However, the data in Fig.4a indicated that feeding *ycbB(-)* bacteria had a much stronger impact on *hsp6* expression than did *bcf-1(-)*, suggesting that only some percentage of the effect of PGN on UPR_{mt} goes through BCF-1. This relatively small effect by *bcf-1(-)* is still very significant but would suggest the presence of a BCF-1-independent effect of PGN on UPR_{mt}. This point, if agreeable by the authors, should be made in the text below line 188. On line 191, the authors mentioned that the work in another paper had indicated the role

of PGN muropeptides on UPR^{mt} through acting as an ATP synthase agonist. Perhaps such direct interactions between PGN and mitochondrial proteins represent at least some of the BCF-1-independent impact of PGN on UPR^{mt}. The authors should discuss this issue to help the reader to understand the difference and relationship between these two pathways.

Besides the UPR^{mt} assay, the authors might also want to have an animal growth assay to compare the effect of feeding *ycbB*(-) bacteria (wild-type worms) and mutating *bcf-1* [feeding K12 and *ycbB*(-) bacteria]. Comparing them under the same condition as in Fig. 4A or Fig. 1D would be helpful regarding the above issue (also mentioned below).

Response:

1) The previous study by Tian and Han (2022) demonstrated that bacterial peptidoglycan muropeptides have a beneficial effect on mitochondrial homeostasis. These muropeptides act as ATP synthase agonists, inhibiting the mitochondrial unfolded protein response (UPR^{mt}). In contrast, a lack of muropeptides, as observed in PG mutant *E. coli* feeding, leads to inhibition of ATP synthase activity, increased ROS production, and activation of UPR^{mt} (Tian and Han, 2022).

In our study, we discovered that PGN interacts with BCF-1. To investigate the role of PGN-BCF-1 interaction in maintaining UPR^{mt}, we added PGN to animals fed with $\Delta ycbB$ (PG mutant *E. coli* lacking specific muropeptides). Interestingly, the addition of PGN suppressed the UPR^{mt} in these animals. However, when PGN was added to *bcf-1* mutant worms, this suppression effect was eliminated (Fig. 4a). This finding suggests that the interaction between PGN and BCF-1 is critical for UPR^{mt} maintenance. However, as pointed out by the reviewer, the impact of feeding *ycbB*(-) bacteria on *hsp-6* expression was much stronger compared to *bcf-1*(-) mutant worms. This suggests that only a portion of the effect of PGN on UPR^{mt} occurs through BCF-1. Although the effect of *bcf-1*(-) is relatively small, it is still significant. These results suggest the presence of a BCF-1-independent effect of PGN on UPR^{mt}. We fully acknowledge the reviewer's suggestion and have included this information in the text.

We added below information in the text:

“We found that supplementation of PGN to animals fed with $\Delta ycbB$ suppressed UPR^{mt}, with this effect being abolished in *bcf-1* mutant worms (Fig. 4a), (Fig. 4a), suggesting that PGN-BCF-1 interaction plays a critical role in maintaining the UPR^{mt}. However, the impact of feeding $\Delta ycbB$ bacteria on *hsp-6* expression was more pronounced compared to that of *bcf-1*(-) mutant worms, indicating a potential BCF-1-independent effect of PGN on UPR^{mt}. Previous research (Tian and Han, 2022) have demonstrated that PGN can inhibit UPR^{mt} through its binding to ATP synthase, leading to increased ATP synthase activity. Thus, there is a possibility of PGN-BCF-1 and PGN-ATP synthase interactions in the regulation of UPR^{mt}”

2) As reviewer's suggestion, we also conducted growth assay under the same condition as in Fig. 4a. We asked if PGN's function in activating food digestion, thereby promoting animals' growth through BCF-1. We found that adding PGN to the animals fed with $\Delta ycbB$ could promote animals' growth, and that growth-promoting effects was eliminated when the PGN was added to *bcf-1* mutant worms (New data in Supplementary Fig. 4), suggesting that PGN-BCF-1 interaction plays a critical role in activating food digestion and promoting animals' growth.

3) As per the reviewer's suggestion, we conducted an animal growth assay. We observed that under the condition of PGN deficiency (HK-*ymcB*+SS), the growth of the animals was significantly slower. In contrast, the presence of PGN-containing HK-*E. coli* promoted the animals to digest SS for growth. Interestingly, we found that HK-*E. coli* was unable to promote the digestion of SS in *bcf-1* mutant animals (New data in Supplementary Fig. 3b). This suggests that HK-*E. coli* promotes digestion through the action of the PGN-binding protein BCF-1.

Moreover, we noticed that the impact of the *bcf-1* mutation on food digestion, in terms of SS digestion, was much stronger compared to the impact of PGN deficiency (*ycbB*-) (New data in Supplementary Fig. 3b). These findings support the notion that there may exist a BCF-1-independent effect of PGN on food digestion. This observation opens up intriguing possibilities for future investigations in this area.

4. Fig. 6B-D. The data for PMK-1 acting downstream of BCF-1 look solid, but the authors did not provide a strong support for UPRmt acting downstream or upstream of PMK-1. In Fig. 6D, while the growth of the *atfs-1(et18)*(increased UPRmt) mutant was improved by mutating *pmk-1*, the growth of the double mutant was lower than that of *pmk-1*(-) alone and the difference in the latter is much greater. From the data, we could also argue that *atfs-1(gf)* suppressed the beneficial effect of *pmk-1*(-) in growth, which could be used to suggest that PMK1 acts upstream of UPR. Because UPRmt and PMK-1 play roles in, or impact, many different signaling pathways, it is not reasonable to make too much out of the animal growth assay that could be affected by multiple pathways. If the authors want to suggest that PMK-1 acts downstream or upstream of UPRmt for digestion and growth, then more molecular markers would be needed for the epistasis analysis.

It is noticeable that the authors did not place PMK-1 in the model in Fig. 7C, as they probably noticed the issue. Since the role of PMK-1 was a key finding in their 2022 paper, it is good that the relationship with PMK-1 was investigated. Besides potentially doing

additional tests to address the issue mentioned above, the authors may make significant modifications to describe the result related to Fig. 6D (line 266-273) and perhaps also discuss the potential relationship in the Discussion.

Response:

To assess the relationship between UPRmt and PMK-1, we firstly examined the level of phosphorylated PMK-1 (p-PMK-1) in *atfs-1 (et18)* mutant animals, which exhibit constitutive activation of UPRmt. Interestingly, we observed an increase in the p-PMK-1 level in *atfs-1 (et18)* mutant animals, suggesting a positive regulatory effect of UPRmt activation on PMK-1 (New data in Fig. 6d). Subsequently, we measured the expression level of *hsp-6*, a marker of UPRmt, and made the following observations: i) The *hsp-6* expression level remained unchanged in *pmk-1* mutant animals, indicating that PMK-1 does not significantly affect UPRmt. ii) Remarkably, the *hsp-6* expression level remained very high in *atfs-1 (et18) pmk-1* double mutant animals (New data in Fig. 6e). This finding suggests that PMK-1 has limited impact on UPRmt regulation.

In the animal growth assay, we found that UPRmt activation inhibits animals' growth in wild-type N2 and *pmk-1* mutant background, suggesting that UPRmt activation inhibits food digestion. However, we also observed that the slow growth phenotype observed in *atfs-1(et18)* mutant animals was partially rescued upon *pmk-1* mutation (New data in Fig. 6f). This suggests that PMK-1 contributes partially to the inhibitory effect of UPRmt activation on food digestion. Furthermore, our findings suggest the involvement of other PMK-1-independent pathways in the inhibition of food digestion under UPRmt activation.

Fig. 6e Results of qRT-PCR analysis showing the expression of *hsp-6* in L4 animals grown under OP50 feeding conditions.

Fig. 6f Developmental progression of *atfs-1(et18)*, *pmk-1(km25)* and *atfs-1(et18);pmk-1(km25)* grown on HK-*E. Coli*+SS at 4d at 20°C. Scale bar, 200µm.

5. The data in Fig. 1A and Fig. 7A seem to be the basis for the authors to propose the interesting hypothesis that PGN-BCF1 signaling is a system that facilitates adaptation of bacteria-eating worms to increase the ability to consume a wide range of food in their natural environment. However, the rationale for the hypothesis does not seem to be sufficient, or it is at least not well explained. In Fig. 1A, it is okay to call HK-*E. coli* low-quality food, based on its ability to support growth, but because it is artificially made (heating) from “high quality food” *E. coli*, the result might not have much power to model

what is found in nature. As for the experiment in Fig. 7A, because only one condition was tested using only a growth assay, the result may raise multiple questions. Are we assuming only *E. coli* can produce the signal? Are we assuming BCF-1 does not interact with anything from other bacterial species? To increase the significance of the test, the authors may want to design additional tests with different combinations. A control with 100% *E. coli* and 0% *E. coli* would need to be part of it. As mentioned earlier, whether the other bacterial species (*E.f* and *B.S*) can produce functional PGN is not clear. These data seem to be important for forming the hypothesis.

At this point, the data seem to support a model that the PGN-BCF-1 interaction acts as a "good-food signal" to promote food uptake and animal growth in bacteria-eating worms. The authors might stay with such a notion, which is already novel and significant, in the abstract and the text. They might discuss additional speculative ideas in the Discussion.

Response:

Based on the reviewer's suggestion, we investigated the effects of PGN from different sources (*E. coli* and *BS*, but not *EF*) on the animals' ability to digest *SS*. Our findings demonstrate that PGN-BCF1 signaling plays a crucial role in facilitating the animals' digestion of inedible food, such as *SS*. In Figure 1, we proposed the hypothesis that low-quality food (*HK-E. coli*) activates animals to consume inedible food, thus increasing the diversity of food sources available to them. This potentially enhances their fitness by maintaining a larger population in their natural environment. Our study provides evidence for the importance of the PGN-BCF1 signaling pathway in enabling animals to digest inedible food, *SS*.

To further investigate whether the PGN-BCF1 signaling system contributes to the adaptation of bacteria-eating worms by enhancing their ability to consume a wide range of food sources in their natural environment, we assessed the population size of animals under different feeding conditions. We initiated the experiments by seeding a single animal in various food conditions and scored the total number of animals (population) after several days of culture. If animals possess a high food digestion ability, they are expected to grow rapidly and exhibit a larger population.

Our results revealed the following observations:

i) Under *SS* feeding conditions, the animals were unable to grow, consequently unable to increase their population. Addition of *E.f*, which cannot activate animals to digest *SS*, did not lead to an increase in the animals' population. In contrast, the addition of high-quality food such as *E. coli* or *B.s* resulted in increased animal populations due to enhanced food digestion (New data in Fig. 7a, b).

ii) When more complex food conditions were introduced, the addition of *B.s* and *E.f* to *SS* resulted in the activation of animals to digest *SS* and subsequent population growth. However, the addition of high-quality food *E. coli* (*OP50*) had the opposite effect in increasing the animal population under *OP50+SS+BS+EF* feeding conditions

(New data in Supplementary Fig. 7a, b). These findings suggest that high-quality food such as *E. coli* promotes animals to digest a wide range of food sources as part of their adaptive strategy.

iii) Under complex feeding conditions (OP50+SS+B.s+E.f), the total number of animals decreased in the *bcf-1* mutant, indicating that the interaction between PGN and BCF1 acts as a "good-food signal" to promote food uptake and animal adaptation in bacteria-eating worms (Fig. 7c).

6. The model figure in Fig. 7C needs significant modification. (1) The authors need to have a better figure legend to describe/explain the arrows and bars, particularly since the model is not specifically described in the Discussion where the figure is cited. If based on the data in the lab's other papers, citations may be made. (2) If the authors agree with the point in 3 above, PGN should have a direct inhibitory bar to UPRmt, which may be mentioned/explained in the legend. (3) It is not clear why the authors have a big blue arrow pointing directly from BCF1 to inedible food. The data in Fig. 4 and 5 seem to support UPRmt acting downstream of BCF1 and it is not clear what data supports a UPRmt-independent function of BCF1. Perhaps the authors were thinking about the epistasis data involving PMK1 that seem to indicate that PMK1 acts downstream of BCF1 but not necessarily downstream of UPRmt (mentioned under 4 above). However, there is

also no clear evidence against the idea that PMK1 is acting upstream of UPRmt (additional tests could clarify this). Without seeing a UPRmt-independent BCF1 activity, this arrow seems unnecessary, regardless of the PMK1 issue. The authors may need to re-think these complicated issues to improve the figure. The figure needs to be made to match the proposal with data and their interpretations. (4) Ideally, PMK-1 should be in the model. If not, the authors may need to mention its known relationship with other factors in the legend.

(5) In the model, "inedible food" is confusing. It might be changed to digestion of inedible bacteria. (6) Is the signaling system also important for digesting high quality food? If yes, you may take out inedible food from the figure (use just "digestion" or "digestion of edible and inedible bacteria").

Mentioned as a minor point below, the linear relationship between BCF1 and NLP3 is suggested by the data but might not be concluded. An alternative way to illustrate this relationship would be to have a curved inhibitory T-bar from BCF-1 to UPRmt, and then write "neuropeptides/NLP-3" next to the arrow to indicate participating or required factors, which can be described in the legend. Potentially, PMK-1 could be put in the same spot of the pathway with the explanation in the legend.

Response: Thanks for reviewer's good suggestions. We changed the model.

Comments on relatively minor issues:

- Fig 1A. on row (d), calling it "low nutrition food" only in this row does not make good sense. The authors should still use "Low Quality Food". Several previous studies have described the micronutrient deficiencies of Heat-killed E. coli. The authors may want to add a sentence in the Results or Introduction to describe these defects to define this food.

Response: We changed to low quality food. We also cite some studies to describe the Heat-killed E. coli.

- Fig. 1C is confusing. A better diagram is needed to illustrate the screen.

Response: We changed

- Fig. 2B and 2C. As mentioned above under #3, it would be good to have a test on worms fed K12 E. coli as in Figure 4a or Fig. 1D so that ycbB(-) feeding could be added as a control (the test in Fig.1D). In this way the extent of the effects under the two conditions can be compared, which may help to address the issue mentioned under 3 above.

Response: Under the condition where PGN is absent (HK-ymcB+SS), we observed

slow growth in animals. The presence of HK-*E. coli* (with PGN) promoted animals to digest SS for growth. However, even in the presence of HK-*E. coli*, the *bcf-1* mutant animals were unable to effectively digest SS, indicating that HK-*E. coli* promotes digestion through the PGN-binding protein BCF-1. Interestingly, the impact of the *bcf-1* mutation on SS digestion was much stronger than the absence of PGN (*ycbB*-), suggesting the presence of a BCF-1-independent effect of PGN on food digestion (New data in Supplementary Fig. 3b). This finding raises intriguing possibilities for future studies.

- Fig. 3. This lab found BCF-1 as a protein binding to *E. coli* in a previous publication, and also mentioned that BCF-1 was separately identified as a PGN-binding protein in another paper. Since *E. coli* binding is involved in bacterial colonization in the worm (He et al paper), such an interaction seems to involve the large/intact PGN molecule on the bacterial surface. Therefore, it is not clear if the bacterial binding discussed in the other publications is the same as the binding shown in this paper. It would be good for the authors to clarify this. By the way, does the binding activity in either case depends on lipoprotein on bacterial membrane? Lipoprotein could be eliminated by certain site-specific proteases.

Response: In our previous study, we discovered that BCF-1 has a direct binding affinity towards *E. coli*, facilitating *E. coli* colonization (He et al., 2023). However, we observed a decrease in BCF-1 binding to bacteria when *ydeR*, a putative fimbrial protein, was mutated in *E. coli*. This suggests that BCF-1 binds to *E. coli* through its fimbriae. The BCF-1-bacterial fimbriae interaction plays a crucial role in promoting colonization, which is distinct from the focus of the current paper. We discuss this at discussion part.

The protein Pal (peptidoglycan-associated lipoprotein) is anchored in the outer membrane (OM) of bacteria and interacts with Tol proteins (Godlewska et al., 2009). Lipoproteins known to attach to PGN is sensitive for trypsin treatment (Braun and Rehn, 1969). In the PGN extraction method, treatment with trypsin is used in the final step of PGN isolation, thus, we used lipoprotein free PGN to do the experiment. In our in vitro PGN-BCF-1 binding experiment, i) we incubated PGN with worm lysate (BCF-1::FLAG) and found that PGN bound to BCF-1 in a concentration-dependent manner (Fig. 3c), suggesting that lipoprotein free PGN interacts with BCF-1. ii) We treated PGN with proteinase K and found that proteinase K-treated PGN cannot activate animals to digest SS (Fig. 1d). Moreover, we found that proteinase K-treated PGN cannot pull-down BCF-1 (New data in Fig. 3d), suggesting that the short amino acid peptides associated with muropeptides are necessary for binding to BCF-1.

- Fig. 3D. PGN addition caused an increased level of BCF-1:GFP. PGN seems to promote the expression of *bcf-1*. Is the regulation at the transcriptional level? It is not common that the presence of signal also regulates the expression of the receptor, which could be interesting if confirmed. What is the potential theory on this? The authors may want to comment on it.

Response: In our study, we utilized BCF-1::GFP::Flag knock-in transgenic animals to investigate the levels of BCF-1 protein under different feeding conditions: SS, SS+PGN, and SS+HK-E. coli. We observed that both PGN and HK-E. coli can induce an increase in BCF-1 protein levels. To examine whether HK-E. coli or E. coli induce *bcf-1* expression at the transcriptional level, we analyzed RNA-seq data obtained from previous research (Liu et al., 2024) (<https://doi.org/10.7554/eLife.94181.1>). This study involved transcriptomics analysis of worms cultured in the absence of food and

under HK-*E. coli* or live *E. coli* feeding conditions. Our analysis revealed that neither HK-*E. coli* nor *E. coli* can induce *bcf-1* mRNA expression (Response Figure-1). These findings suggest that *E. coli* induces the production of BCF-1 at the protein level. It is plausible that the overall translation of proteins is increased following *E. coli* consumption, leading to elevated BCF-1 protein levels and subsequently promoting food uptake.

- Fig. 5E, the test was done using only the *bcf-1*(-) mutant worms, which is effective to show the rescue of *bcf-1*(-) defect in growth. However, it would be helpful to have the wild-type worms as a control so that the extent of the rescue is seen. Is the ~30-50% increase in body length sufficient to suppress most or only part of the defect? This would not take away the suppression role but may help to understand the *bcf-1* roles and the pathway.

Response:

As per reviewer's suggestion, we added the wild-type animal as control. On HK-*E. coli*+SS feeding condition, we observed digestion defects and developmental delay in *bcf-1* (*ok2599*) mutants. However, we found that these defects were partially restored when we performed RNA interference (RNAi) experiments targeting *nlp-3*, *egl-3*, and *egl-21* (New data in Fig. 5e). Although this partial rescue indicated the involvement of *nlp-3*, there were still residual defects observed, suggesting the presence of other factors, in addition to *nlp-3*, that play a role in BCF-1-mediated regulation of food

digestion.

- Similarly, for Fig 6C and D, having a wild-type worm control would help to learn the extent of suppression by *pmk-1(-)*, that would help to view its function in the pathway.

Response: We add the wild-type animals as control.

In our animal's growth assay, we observed a complete recovery of the developmental delay phenotype in *bcf-1* mutant animals upon mutation of *pmk-1* (New data in Fig. 6c). Additionally, we found that the levels of phosphorylated PMK-1 (p-PMK-1) were increased in the *bcf-1* mutant. Based on these findings, it can be inferred that the mutation of *bcf-1* induces innate immunity, which in turn inhibits food digestion and leads to the observed developmental delay phenotype.

In the animal growth assay, we found that UPRmt activation inhibits animals' growth in wild-type N2 and *pmk-1* mutant background, suggesting that UPRmt activation inhibits food digestion. However, we also observed that the slow growth phenotype observed in *atfs-1(et18)* mutant animals was partially rescued upon *pmk-1* mutation (New data in Fig. 6f). This suggests that PMK-1 contributes partially to the inhibitory effect of UPRmt activation on food digestion. Furthermore, our findings suggest the involvement of other PMK-1-independent pathways in the inhibition of food digestion under UPRmt activation.

- In Fig. 5, the data supporting the roles of *nlp-3*, *egl-3*, and *elg-12* on UPRmt are solid. The suppression data by RNAi of these genes in the *bcf-1(-)* mutants provided further support for UPRmt acting downstream of BCF-1. It is also reasonable to state, as the

authors did, that the suppression data suggested that *nlp-3* acts downstream of BCF-1, even though the extent of suppression is not clear based on the data in Fig. 5D&E (lack of positive control such as *bcf1(wt)* animals). However, there is no evidence that the 1.43-fold expression increase (Fig. S4B) would be the driver of the phenotype. Contributions could also be made by other NLPs. At this point, the safe thing to say is that the *bcf-1(-)* impact on URPmt at least partially depends on NLP-3. The T-bar in the model from BCF-1 to NLP-3 is a suggestive action.

Response:

1) As per reviewer's suggestion, we added the wild-type animal as control. On HK-E. coli+SS feeding condition, we observed digestion defects and developmental delay in *bcf-1 (ok2599)* mutants. However, we found that these defects were partially restored when we performed RNA interference (RNAi) experiments targeting *nlp-3*, *egl-3*, and *egl-21* (New data in Fig. 5e). Although this partial rescue indicated the involvement of *nlp-3*, there were still residual defects observed, suggesting the presence of other factors, in addition to *nlp-3*, that play a role in BCF-1-mediated regulation of food digestion.

2) As per reviewer's suggestion, we also added the wild-type animal as control. We found that *bcf-1* mutation induced UPR^{mt} was all most inhibited by knocking down *nlp-3* (New data in Fig. 5d).

- Line 244. Figure S5 should be Figure S4.

Response: changed

- Fig. 6A: a non-phosphorylated PMK-1 level for loading control would be better than the tubulin control to evaluate the significance of the change seen in p-PMK-1.

Response: We add the non-phosphorylated PMK-1 level.

- Sample size. in this study, the sample sizes in some experiments seem quite low. In some cases, the legends indicate that N= the number of animals scored from at least three independent experiments, which would mean even smaller number of worms were scored per condition/experiment. These numbers are low for a worm study. The data and conclusions can only be improved by increasing the sample sizes.

Response: We increased sample size.

Reviewer #2 (Remarks to the Author):

Review on the manuscript entitled „Bacterial peptidoglycan as a food digestive signal that facilitates adaptation of animals in nature” by Qi and colleagues. The authors used the nematode *Caenorhabditis elegans* as a tractable genetic model feeding on various bacteria (mainly *Escherichia coli*) and other microorganisms (so, consumes a wide range of foods) to demonstrate that bacterial peptidoglycan (PGN) produced by *E. coli* cells is a food signal for the host to digest other, normally inedible food resources (*C. elegans* does

not grow on *Staphylococcus saprophyticus* bacteria, but can do when it is also exposed to a low-quality food resource such as heat-killed *E. coli* cells). They showed that the host glycosylated protein BCF-1 (bacterial colonization factor) directly interacts with the bacterial PGN in the gut to promote the release of NLP-3 (neuropeptide-like protein) neuropeptide, which eventually inhibits the mitochondrial UPRmt (unfolded protein response) in the gut cells (a cell non-autonomous mechanism). Constitutively active UPRmt indeed prevents food digestion in an innate immunity protein PMK-1-dependent manner. These are very important results with potent human implications (highlighting the importance of the human microbiome in adaptation and survival). Experiments are well-designed and data are solid. However, it would be important to show that this phenomenon is evolutionarily conserved to higher taxa. To publish the study in *Nat Commun*, I guess it would be important to show that the underlying molecular machinery may also present in higher animal taxa (mammals) despite the fact that some of the key components (e.g. BCF-1 orthologs) are missing from higher organisms. So, the authors should demonstrate the evolutionary conservation of the phenomenon they discovered here or show that it is a nematode-specific trait. Thus, I propose some additional experiments to complete successfully the story.

Response: Thanks for the reviewer's kindly suggestions and comments which are very helpful for our study.

Major comments:

1, The authors should demonstrate that *E. coli* PGN can (or cannot) influence intestinal UPRmt in mammals (humans or mice).

Response:

In a previous study conducted by Min Han's lab (Tian and Han, 2022) (Developmental Cell 2022, PMID: PMC8825754), it was discovered that bacterial peptidoglycan muropeptides play a beneficial role in mitochondrial homeostasis and animal physiology by acting as ATP synthase agonists in *C. elegans*. Similarly, Schwarzer et al (Schwarzer et al., 2023) (Science 2023, DOI: 10.1126/science.ade976) observed that bacterial cell walls or purified NOD2 ligands stimulate growth in mice. However, the specific mechanism by which bacterial cell walls promote animal growth through the regulation of food digestion remains unknown.

In our current study, we found that peptidoglycan (PGN) is sensed by the conserved intestinal glycosylated protein BCF-1, leading to the inhibition of the mitochondrial unfolded protein response (UPRmt) through the action of the neuropeptide NLP-3, which in turn promotes food digestion. Since *E. coli* is also present in the gut of mammals, it is plausible that PGN could impact the intestinal UPRmt in mammals as well.

Interestingly, recent work from Han's lab (Tian et al., 2023) (biorxiv 2023, doi:<https://doi.org/10.1101/2023.01.05.522895>) has demonstrated that bacterial

muropeptides promote oxidative phosphorylation (OXPHOS) and suppress mitochondrial stress in both normal and human mitochondrial disease models. Their findings showed that *E. coli*-derived PG muropeptides accumulate within mitochondria of mouse intestinal epithelial cells (IECs) and significantly reduce the expression of the mitochondrial stress marker hsp60 in human mitochondrial disease models, specifically LS mutant fibroblast cells isolated from a patient with Leigh Syndrome and hypertrophic cardiomyopathy (GM13411, Coriell Institute). This study strongly suggests that *E. coli* PGN is also involved in suppressing mitochondrial stress in mammals. In our current paper, we cite this research and discuss its implications for further reinforcing the evolutionarily conserved function of PGN in suppressing the UPRmt.

2, They should investigate whether a PGN mutant diet inducing UPRmt can result in increased food avoidance behavior (in mice).

Response:

In our previous study (Geng et al., 2022), we demonstrated that Heat-killed *E. coli* (HK-*E. coli*) induces animals to digest inedible food *Staphylococcus saprophyticus* (SS). Building upon this, we screened *E. coli* mutants and observed significantly slower worm growth when feeding HK-*E. coli* mutants (*ycbB* and *ygeR*) in combination with SS (Fig. 1d). This suggests that the ability to digest food is reduced in animals when individually fed these HK-mutant *E. coli*. Furthermore, we discovered that peptidoglycan (PGN) interacts with the host protein BCF-1, leading to the inhibition of the mitochondrial unfolded protein response (UPRmt) through the action of the neuropeptide NLP-3, which promotes food digestion in *C. elegans*.

To investigate the impact of *E. coli* PGN on host digestion in mice, an ideal experiment would involve using germ-free or antibiotic-treated mice (to deplete the intestinal microbiota) and supplementing them with or without PGN to assess their food digestion ability.

Recently, in Han lab, they also tested the function of *E. coli* muropeptides derived from PGN in the intestines of mice (Tian et al., 2023) (biorxiv 2023, doi:<https://doi.org/10.1101/2023.01.05.522895>). Firstly, they investigated the role of muropeptides in the small intestine of mice by depleting the intestinal microbiota using an antibiotic cocktail (AIMD). This procedure resulted in germ-free-like phenotypes, including an enlarged cecum and reduced spleen size in both male and female mice. Secondly, since the small intestine is known as the primary site for nutrient sensing and absorption, they monitored food consumption and stool output in AIMD-treated mice to assess the modulation of nutrient absorption by muropeptides. While neither AIMD nor PG feeding caused significant changes in food consumption within 24 hours, AIMD-treated mice produced more stool during this period,

indicating compromised digestion and nutrient uptake due to microbiota depletion. Importantly, oral gavage of PG partially but significantly suppressed the increase in stool output in male mice, suggesting that PG muropeptides promote nutrient absorption by regulating mitochondrial metabolism and maintaining small intestinal homeostasis. Lastly, consistent with these findings, AIMD treatment led to suppressed weight gain in mice, which was partially rescued by oral gavage of PG in male mice.

Collectively, these data provide support for the notion that muropeptides promote nutrient absorption in the small intestine of mice. In our current paper, we cite this research and discuss its implications for further reinforcing the evolutionarily conserved function of PGN in promoting digestion.

3, It should be discussed which bacterial taxa contain PGN (does *S. saprophyticus* contain or not, and if yes, why it is ineffective in this respect).

Response:

All bacteria possess cell walls, and one major component of these cell walls is bacterial peptidoglycan. Peptidoglycan is a polymer composed of sugars and amino acids, with N-acetylglucosamine (NAG) and N-acetylmuramic acid (NAM) as the primary sugars joined by glycosidic bonds. Peptide bonds connect the amino acids to the NAM sugars. The structure of peptidoglycan is responsible for providing shape, rigidity, and strength to the bacterial cell wall. Maintaining shape is crucial for bacterial survival in various environments. Additionally, peptidoglycan is involved in cell division, facilitating bacterial reproduction.

Schwarzer et al. (Schwarzer et al., 2023) (Science 2023, DOI: 10.1126/science.ade976) discovered that *Lactiplantibacillus plantarum* (strain Lp^{WJL}) stimulates the growth of mice. They identified cell walls isolated from Lp^{WJL}, as well as ligands of the pattern recognition receptor NOD2, as sufficient bacterial cues for promoting animal growth. Their study demonstrated that Lp^{WJL} peptidoglycans interact with NOD2 in the intestinal epithelium, leading to increased postnatal growth despite undernutrition. However, another strain, Lp^{NIZO2877}, had no impact on growth, suggesting that the growth-promoting effects of Lp^{WJL} are strain-specific.

In our research, we observed that peptidoglycan from *E. coli* and *B.s* can promote animal digestion of SS for growth, whereas peptidoglycan from *E. faecalis* does not have the same effect. Therefore, the findings of Schwarzer et al. and our own studies imply that the effects of bacterial peptidoglycan on growth may be strain-specific.

4, It is clear that inhibiting the UPRmt leads to food diversification in this organism. However, what is the biology behind this phenomenon, what is the function of UPRmt/innate immune pathway in this response (why induced UPRmt results in increased food avoidance behavior)? How does UPRmt induces the innate immune pmk-1 pathway and what is the role of the pathway in food selection?

Response:

In our study, we discovered that activation of the mitochondrial unfolded protein response (UPRmt) leads to the inhibition of food digestion and utilization in animals. The UPRmt is a transcriptional response activated by various forms of mitochondrial dysfunction and regulated through communication between the mitochondria and nucleus (Shpilka and Haynes, 2018). UPRmt activation plays a role in repairing and restoring mitochondrial function to maintain cellular homeostasis.

At the physiological level, when the UPRmt is activated, animals tend to shut down food digestion and avoid consuming food. This response serves as a mechanism to reduce protein translation and cellular metabolism, resembling a state of starvation, which promotes recovery and alleviation of cellular stress. This biology function of UPRmt activation in regulating food digestion and behavior is an important aspect we investigated in this study.

To further understand the underlying mechanism, we conducted experiments to test the dependence of UPRmt activation-induced food avoidance on specific factors. Firstly, we performed food avoidance experiments in which we knocked down *nlp-3*, a gene involved in UPRmt activation, in animals with a mutation in *bcf-1*. We observed that knocking down *nlp-3* reduced the food avoidance response induced by the *bcf-1* mutation (New data in Supplementary Fig. 5c), indicating that the food avoidance behavior in *bcf-1* mutants is dependent on UPRmt activation.

Furthermore, we conducted food avoidance experiments using animals with a double mutation in *bcf-1* and *pmk-1*. We found that the food avoidance response induced by the *bcf-1* mutation was diminished in the *bcf-1*, *pmk-1* double mutants (New data in Supplementary Fig. 6), indicating that the food avoidance behavior in *bcf-1* mutants indeed depends on *pmk-1*.

These findings provide insights into the molecular and genetic basis of the UPRmt-induced food avoidance response and its dependency on specific signaling pathways.

Minor points:

1, The text, mainly the abstract, should be polished throughout.

Response: changed.

2, Title: my suggestion: Bacterial peptidoglycan as a food digestive signal that facilitates

host adaptation to consuming more diverse food resources.

Response: We agreed with this title.

3, Abstract: abbreviations should be shown in detail! BCF-1 (bacterial colonization factor), UPR (unfolded protein response), NLP-3 ..., PMK-1, ...

My suggestion: Food availability and usage is a major adaptive force for the successful survival of animals in nature. However, very little is known about the signal from food to activate the hosts digestive system, which facilitates animals to digest more diverse food in nature.

change to:

Food availability and usage is a major adaptive force for the successful survival of animals in nature. Despite its physiological and medical importance, very little is known about the food signal that activates the host digestive system to facilitate the consumption of more diverse foods in nature.

„We identified that a glycosylated protein BCF-1 in the gut that interacts ...” change to

„We identified that the host’s glycosylated protein ... (BCF-1) in the gut interacts ...”

Text:

Response: changed.

4, Fig. 1 (and also in the relevant part of the text): SS should be written „S. saprophyticus” or at least “S. saproph.”

Response: In our text, we have already used the abbreviation "SS" to refer to *Staphylococcus saprophyticus*. While we understand that "S. saproph." is another possible abbreviation, it may not be feasible to use it in the figure panel since it could be cumbersome.

5, line 131: “the slow growth phenotype of worms fed SS” should be changed to “the slow growth phenotype of worms fed on SS”

Response: changed.

6, On some figure panels, E. Coli should be changed to E. coli

Response: changed.

7, line 229: the classical pathway of UPRmt should be changed to the canonical pathway of UPRmt

Response: changed.

8, line 263: "(developmental delay) in *bcf-1* mutant was recovered in double mutant (Figure 6B). This should be Figure 6C.

Response: changed.

9, Line 267: "by continuously activation of UPRmt in" should be "continuous activation of UPRmt"

Response: changed.

10, line 271: digestion in *pmk-1* mutant was also inhibit by – rewrite the sentence

Response: changed.

11, line 338: "nematodes could evolute a mechanism" should be "nematodes could evolve a mechanism"

Response: changed.

Reference:

Braun, V., and Rehn, K. (1969). Chemical characterization, spatial distribution and function of a lipoprotein (murein-lipoprotein) of the *E. coli* cell wall. The specific effect of trypsin on the membrane structure. *European journal of biochemistry* *10*, 426-438.

Geng, S., Li, Q., Zhou, X., Zheng, J., Liu, H., Zeng, J., Yang, R., Fu, H., Hao, F., Feng, Q., *et al.* (2022). Gut commensal *E. coli* outer membrane proteins activate the host food digestive system through neural-immune communication. *Cell Host Microbe* *30*, 1401-1416. e1408.

Godlewska, R., Wiśniewska, K., Pietras, Z., and Jagusztyn-Krynicka, E.K. (2009). Peptidoglycan-associated lipoprotein (Pal) of Gram-negative bacteria: function, structure, role in pathogenesis and potential application in immunoprophylaxis. *FEMS microbiology letters* *298*, 1-11.

He, Y., Hao, F., Fu, H., Tian, G., Zhang, Y., Fu, K., and Qi, B. (2023). N-glycosylated intestinal protein BCF-1 shapes microbial colonization by binding bacteria via its fimbrial protein. *Cell Rep* *42*, 111993.

Liu, P., Liu, X., and Qi, B. (2024). UPRER–immunity axis acts as physiological food evaluation system that promotes aversion behavior in sensing low-quality food. *Elife*, 13:RP94181.

Schwarzer, M., Gautam, U.K., Makki, K., Lambert, A., Brabec, T., Joly, A., Srutková, D., Poinot, P., Novotná, T., Geoffroy, S., *et al.* (2023). Microbe-mediated intestinal NOD2 stimulation improves linear growth of undernourished infant mice. *Science* *379*.

Shpilka, T., and Haynes, C.M. (2018). The mitochondrial UPR: mechanisms, physiological functions and implications in ageing. *Nat Rev Mol Cell Biol* *19*, 109-120.

Tian, D., Cui, M., and Han, M. (2023). Bacterial muropeptides promote OXPHOS and suppress

mitochondrial stress in normal and human mitochondrial disease models. bioRxiv, 2023.2001.2005.522895.

Tian, D., and Han, M. (2022). Bacterial peptidoglycan muropeptides benefit mitochondrial homeostasis and animal physiology by acting as ATP synthase agonists. *Dev Cell*.

REVIEWERS' COMMENTS

Reviewer #1 (Remarks to the Author):

The authors made strong efforts in addressing essentially all my comments including the comments on major issues and minor issues. Many additional experiments were done in these efforts. The manuscript has been significantly improved and the acceptance of publication is thus recommended.

The original manuscript was criticized for having too many writing errors. The revision is significantly improved in this regard. However, the authors/editors may make additional effort in this regard before publication.

Reviewer #2 (Remarks to the Author):

In this study the authors show that bacterial (*E. coli*) peptidoglycan (PGN) is sensed by the *C. elegans* intestinal glycosylated protein BCF-1, leading to the inhibition of the mitochondrial unfolded protein response (UPR_{mt}) through the action of the neuropeptide NLP-3. Repressed UPR_{mt} in turn promotes food digestion in the host.

Upon the referees' requests, the revision contains many novel data that made the story stronger.

Although I saw that this work is basically a *C. elegans*-based study, my main critics dealt with the evolutionarily conservation of the phenomenon which is a key issue.

In their response, the authors say that „Since *E. coli* is also present in the gut of mammals, it is plausible that PGN could impact the intestinal UPR_{mt} in mammals as well.” This is that the authors should actually show instead of discussing on this issue. This was my main requirement. I request it again, using a tractable mammalian system. So, in this study the authors should experimentally demonstrate that *E. coli* PGN is also involved (or not) in suppressing mitochondrial stress in a mammalian system. Using normal (control) vs. PGN defective *E. coli* cells may can address this issue (treatments should be applied after a strong antibiotic treatment).

I guess, it is not a good strategy to cite biorxiv publications because many of them have not yet undergone a reviewing process, and still remain unpublished in peer-reviewed papers for a certain amount of time.

REVIEWERS' COMMENTS

Reviewer #1 (Remarks to the Author):

The authors made strong efforts in addressing essentially all my comments including the comments on major issues and minor issues. Many additional experiments were done in these efforts. The manuscript has been significantly improved and the acceptance of publication is thus recommended.

The original manuscript was criticized for having too many writing errors. The revision is significantly improved in this regard. However, the authors/editors may make additional effort in this regard before publication.

Response:

We appreciate Reviewer-1 for taking the time to provide valuable feedback on our paper. Based on the reviewer's suggestions, we have made revisions to address writing errors and improve the language used in the manuscript. Thank you for helping us to enhance the quality of our work.

Reviewer #2 (Remarks to the Author):

In this study the authors show that bacterial (*E. coli*) peptidoglycan (PGN) is sensed by the *C. elegans* intestinal glycosylated protein BCF-1, leading to the inhibition of the mitochondrial unfolded protein response (UPR_{mt}) through the action of the neuropeptide NLP-3. Repressed UPR_{mt} in turn promotes food digestion in the host.

Upon the referees' requests, the revision contains many novel data that made the story stronger.

Although I saw that this work is basically a *C. elegans*-based study, my main critics dealt with the evolutionarily conservation of the phenomenon which is a key issue.

In their response, the authors say that „Since *E. coli* is also present in the gut of mammals, it is plausible that PGN could impact the intestinal UPR_{mt} in mammals as well.” This is that the authors should actually show instead of discussing on this issue. This was my main requirement. I request it again, using a tractable mammalian system. So, in this study the authors should experimentally demonstrate that *E. coli* PGN is also involved (or not) in suppressing mitochondrial stress in a mammalian system. Using normal (control) vs. PGN defective *E. coli* cells may can address this issue (treatments should be applied after a strong antibiotic treatment).

I guess, it is not a good strategy to cite biorxiv publications because many of them have

not yet undergone a reviewing process, and still remain unpublished in peer-reviewed papers for a certain amount of time.

Response: We are grateful for the valuable suggestions and comments provided by the reviewer, which have greatly contributed to the improvement of our study.

So, in this study the authors should experimentally demonstrate that E. coli PGN is also involved (or not) in suppressing mitochondrial stress in a mammalian system.

Response: I totally agree with the reviewer's suggestion to investigate the evolutionary conservation of PGN's role in inhibiting mitochondrial stress using a mammalian system.

Previously, researchers in Min Han's lab at the University of Colorado Boulder discovered that bacterial peptidoglycan muropeptides play a role in maintaining mitochondrial homeostasis and animal physiology by acting as ATP synthase agonists in *C. elegans* (Developmental Cell, PMID: PMC8825754). More recently, they have also found that *bacterial muropeptides promote OXPHOS and suppress mitochondrial stress in normal and human mitochondrial disease models* (biorxiv, doi:<https://doi.org/10.1101/2023.01.05.522895>). Their studies demonstrated that (i) *E. coli*-derived peptidoglycan muropeptides accumulated significantly in the mitochondria of mouse intestinal epithelial cells (IECs), and (ii) treatment with *E. coli*-derived muropeptides led to a significant reduction in the expression of the mitochondrial stress marker hsp60 in human mitochondrial disease models, specifically LS mutant fibroblast cells (GM13411, Coriell Institute) derived from a patient diagnosed with Leigh Syndrome (LS) and hypertrophic cardiomyopathy.

Therefore, their study indicates that *E. coli* peptidoglycan (PGN) also has the ability to suppress mitochondrial stress in mammalian systems. **We have cited and discussed their research in our paper to reinforce the evolutionarily conserved role of PGN in suppressing the mitochondrial unfolded protein response (UPR^{mt}).**

I guess, it is not a good strategy to cite biorxiv publications because many of them have not yet undergone a reviewing process, and still remain unpublished in peer-reviewed papers for a certain amount of time.

Response:

I completely understand the reviewer's concerns regarding citing research from bioRxiv. In today's world, the time from submission to publication of a research paper can be lengthy. Preprint servers facilitate the dissemination of scientific information and enable the sharing of the latest discoveries in science.

It is worth noting that reputable preprint servers are accepted for citation by prestigious publications such as Nature Press.

Please See below information which extract from Nature website:

“ <https://www.nature.com/nature/for-authors/formatting-guide>”

References

Only articles that have been published or accepted by a named publication, or that have been uploaded to a recognized preprint server (for example, arXiv, bioRxiv), should be in the reference list; papers in preparation should be mentioned in the text with a list of authors (or initials if any of the authors are co-authors of the present contribution). ”

Therefore, I believe it would be beneficial to consider citing research from bioRxiv for our own work, as this can save time and resources if a study has already been conducted on a particular topic.